# Natural variation in *CTB4a* enhances rice adaptation to cold habitats

Zhanying Zhang[1,*], Jinjie Li[1,*], Yinghua Pan[1,2,*], Jilong Li[1,*], Lei zhou[1,3,*], Hongli Shi[1,*], Yawen Zeng[4], Haifeng Guo[1], Shuming Yang[4], Weiwei Zheng[1], Jianping Yu[1], Xingming Sun[1], Gangling Li[1], Yanglin Ding[5], Liang Ma[5], Shiquan Shen[4], Luyuan Dai[4], Hongliang Zhang[1], Shuhua Yang[5], Yan Guo[5] & Zichao Li[1]

Low temperature is a major factor limiting rice productivity and geographical distribution. Improved cold tolerance and expanded cultivation to high-altitude or high-latitude regions would help meet growing rice demand. Here we explored a QTL for cold tolerance and cloned the gene, *CTB4a* (cold tolerance at booting stage), encoding a conserved leucine-rich repeat receptor-like kinase. We show that different *CTB4a* alleles confer distinct levels of cold tolerance and selection for variation in the *CTB4a* promoter region has occurred on the basis of environmental temperature. The newly generated cold-tolerant haplotype *Tej*-Hap-KMXBG was retained by artificial selection during temperate *japonica* evolution in cold habitats for low-temperature acclimation. Moreover, CTB4a interacts with AtpB, a beta subunit of ATP synthase. Upregulation of *CTB4a* correlates with increased ATP synthase activity, ATP content, enhanced seed setting and improved yield under cold stress conditions. These findings suggest strategies to improve cold tolerance in crop plants.

[1] Key Laboratory of Crop Heterosis and Utilization, Ministry of Education/Beijing Key Laboratory of Crop Genetic Improvement, Department of Plant Genetics and Breeding, China Agricultural University, Beijing 100193, China. [2] Rice Research Institute, Guangxi Academy of Agricultural Sciences, Nanning 530007, China. [3] Hubei Key Laboratory of Food Crop Germplasm and Genetic Improvement, Food Crops Institute, Hubei Academy of Agricultural Sciences, Wuhan 430064, China. [4] Biotechnology and Genetic Resources Institute, Yunnan Academy of Agricultural Sciences, Kunming 650205, China. [5] State Key Laboratory of Plant Physiology and Biochemistry, College of Biological Sciences, China Agricultural University, Beijing 100193, China. * These authors contributed equally to this work. Correspondence and requests for materials should be addressed to Z.L. (email: lizichao@cau.edu.cn).

Rice (*Oryza sativa* L.) is a staple food crop feeding over one-half of the world population and includes two subspecies, *indica* and *japonica*, that originate from tropical or subtropical areas[1,2]. As rice is a cold-sensitive crop, low temperatures restricts its cultivation in cold habitats[2]. Compared with *indica*, *japonica* is mainly planted in higher altitude and latitude habitats. Accordingly, *japonica* rice is more cold tolerant than *indica*[2,3].

Cold injury in rice affects both the vegetative (germination and seedling) and reproductive (booting and flowering) growth stages. Cold stress at the booting stage is a critical factor in rice production as it causes spikelet sterility and increased susceptibility to certain diseases[2,4]. It is a major problem for rice cultivation in 24 countries, including China, Japan and Korea[5]. Rice is widely planted in China, from Hainan island (N: 19–20°) to Mohe River (N: 52–53°) in Heilongjiang, and from the eastern coastal areas to the Yunnan-Guizhou Plateau. China's annual loss of rice due to low temperatures was 3–5 million tons[6]. Therefore, improving cold tolerance at the booting stage is one of the most important tasks for rice breeders and it would be beneficial to develop cold-tolerant rice varieties using genes derived from existing germplasm resources.

Cold tolerance is a complex trait that is controlled by multiple loci and affected by the environment. Compared with other agronomic traits, dissecting the genetic basis of cold tolerance in rice has occurred relatively slowly. Cold tolerance of rice at different growth stages is probably controlled by different genes[3]. In the past decades, although many QTLs conferring cold tolerance at the vegetative or reproductive stage have been mapped on almost 12 chromosomes[2,4,7–20], only a few genes conferring cold tolerance at the vegetative growth stage have been isolated, such as *COLD1*, *qLTG3-1* and *LTG1* (refs 21–23). Only one gene *Ctb1* (ref. 24) controlling cold tolerance at the booting stage has been cloned, and little is known about the underlying molecular mechanisms of cold tolerance at the booting stage, in part because of difficulties in accurately evaluating phenotypes and due to the complexity of the associated genetic pathways.

Yunnan province is located on the southwest plateau of China and is characterized by low average temperatures during the rice growing period (Supplementary Fig. 1a). In our previous work, many cold-tolerant landraces were identified, of which *japonica* variety KUNMINGXIAOBAIGU (KMXBG) was relatively tolerant of cold at both the seedling and booting stages[25,26]. To investigate the genetic basis of cold tolerance at the booting stage, a cross between KMXBG (cold-tolerant) and Towada (cold-sensitive) was made to develop a set of cold-tolerant near-isogenic lines using KMXBG as a donor. Eight QTLs conferring cold tolerance at the booting stage were mapped on chromosomes 1, 4, 5, 10 and 11 (ref. 9).

In this study, we characterized a near-isogenic line (NIL) NIL1913 that exhibited more cold tolerance than the recurrent parent Towada and included the QTL, *qCTB4-1*. We established an advanced back-cross population $BC_6F_2$, fine mapped and cloned the target gene, *CTB4a*, which encodes a leucine-rich repeat receptor-like protein kinase (LRR-RLK). This work uncovers a novel gene for cold tolerance with potential value in breeding.

## Results

### Characterization of NIL1913 and identification of CTB4a.
In our previous work, two linked QTLs, *qCTB4-1* and *qCTB4-2*, were mapped on the short arm of chromosome 4. We constructed a NIL, NIL1913 with *qCTB4-1*, hereafter referred as *CTB4a*, from KMXBG in a Towada genetic background. NIL1913 showed a similar morphology to Towada under normal cultivation conditions in Beijing (Fig. 1a,b and Supplementary Table 1), but had stable improved cold tolerance at the booting stage with higher seed setting and pollen fertility (Fig. 1a–d, Supplementary Table 1 and Supplementary Fig. 2) when subjected to different kinds of cold stress conditions, including cold stress in deep water (CS-DW, Supplementary Fig. 1b,c), in a phytotron (CS-PT, Supplementary Fig. 1b) and in a high-altitude area (CS-HAA, altitude 1,974 m) with naturally low temperatures near Kunming (Supplementary Fig. 1a,b).

To isolate *CTB4a*, we carried out high resolution mapping using 3,102 $F_2$ individuals generated from backcrossing NIL1913 with Towada. Among the 3,102 individuals, a total of 46 recombinants between RM2146 and RM6770 were selected for further work. According to the genotypic groups of markers RM2146 and RM6770, five kinds of recombinant individuals (G1-G5) were grouped (Supplementary Fig. 3a). To eliminate the effect of *qCTB4-2*, 25 recombinant individuals from groups G1 and G2 were analysed. The 25 recombinant individuals were divided into four groups (NA, NB, NC and ND) using RM16349, RM5414 and SSR29 (Supplementary Fig. 3b). There was no significant difference in seed setting between groups NA and NB or between groups NC and ND. However, the seed setting of groups NA and NB were much lower than that of NC and ND (Supplementary Fig. 3b). These results indicated that *qCTB4-1* was located in the region between RM16349 and SSR29. Next, we developed new polymorphic markers for detecting the genotypes of total six recombinant individuals of groups NB and NC, and finally delimited the *CTB4a*-containing DNA fragment to a 56.8 kb region between markers STS5 and STS12 (Fig. 1e).

Within this genomic region, four ORFs were predicted by RGAP (Rice Genome Annotation Project; Fig. 1e). We identified the full-length cDNA corresponding to ORF4 (*LOC_Os04g04330*), which encodes a LRR-RLK (Supplementary Fig. 4a). Sequence analysis showed 9 SNPs and 1 Indel in the promoter region, and 6 SNPs and 1 Indel in the coding region of ORF4 between KMXBG and Towada, among which 2 SNPs (T425C, T2063C) and 1 Indel (CTC 49–51) resulted in amino acid changes (Fig. 1e). The expression of *CTB4a* was highly induced in panicles and leaves of NIL1913 by cold stress compared to Towada (Fig. 1f and Supplementary Fig. 5). We postulated that ORF4 was the probable candidate gene for *CTB4a*.

### CTB4a confers cold tolerance.
To confirm our prediction, a complementary construct containing a 3.2 kb promoter fragment fused with the coding region and 3′-UTR of *CTB4a* from KMXBG was introduced into Towada (Fig. 2a). Phenotypic analysis of two transgenic plants, C11 and C12, with higher *CTB4a* expression (Supplementary Fig. 6a), showed that the cold sensitivity of Towada was rescued. Relative seed setting of C11 and C12 was significantly higher than that of wild type, Towada, under cold stress (Fig. 2b). We concluded that *ORF4* was the functional gene for *CTB4a*.

To determine whether differences in the promoter or coding regions were responsible for the function of *CTB4a*, several experiments were conducted. First, a vector containing pCTB4a$^{KMXBG}$::GUS or pCTB4a$^{Towada}$::GUS (with the GUS reporter gene driven by the promoter of the *CTB4a* allele from KMXBG or the allele from Towada) was introduced into Nipponbare (Nip). Expression analysis showed that the promoter of the *CTB4a* allele from KMXBG showed higher GUS activity than that from Towada under cold stress at the booting stage (Supplementary Fig. 7). Then two overexpression constructs containing the *CTB4a* allele from KMXBG (K-OE) or from Towada (T-OE) driven by the 35 S promoter from tobacco cauliflower mosaic virus (CaMV35S) were separately introduced into Towada (Fig. 2a). The K-OE1 and K-OE12 lines showed

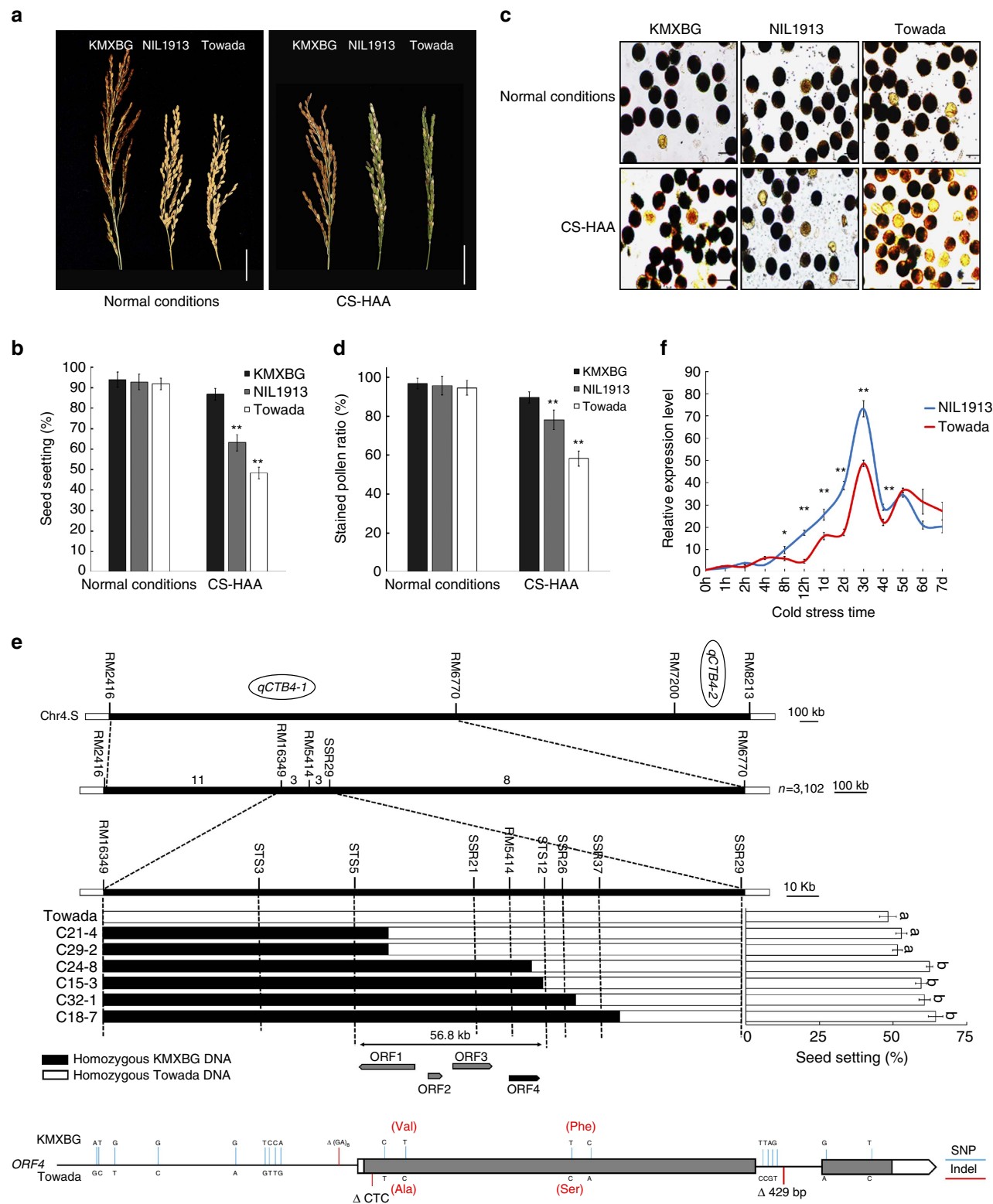

**Figure 1 | Characterization of NIL1913 and identification of *CTB4a*. (a,b)** Panicles (**a**) and seed setting (**b**) of KMXBG, Towada and NIL1913 planted under normal conditions (Beijing) and cold stress at a higher-altitude area (CS-HAA, Kunming). Data represent means ± s.d. ($n = 15$), \*\**P* < 0.01, \**P* < 0.05, Student's *t*-test. Scale bars, 4 cm. (**c,d**) Pollen fertilities evaluated by $I_2$–KI staining (**c**), and the blue stained pollen grains were counted for calculating pollen fertility for samples under CS-HAA (**d**). Scale bars, 25 μm. Data represent means ± s.d. ($n = 10$). (**e**) Fine mapping of *CTB4a*. Left, high resolution mapping. Right, phenotypes of homozygous recombinants. Significance was determined by one-way ANOVA. Schematic of the gene structure and allelic variation in *CTB4a* between KMXBG and Towada indicated by vertical lines at bottom. The presence of the same lowercase letter above the error bar denotes a non-significant difference between the means (*P* > 0.05, Student's *t*-test). (**f**) qRT-PCR analysis using total RNA isolated from panicles of NIL1913 and Towada under CS-PT at different time points. Data represent means ± s.d. ($n = 3$), \*\**P* < 0.01, \**P* < 0.05, Student's *t*-test.

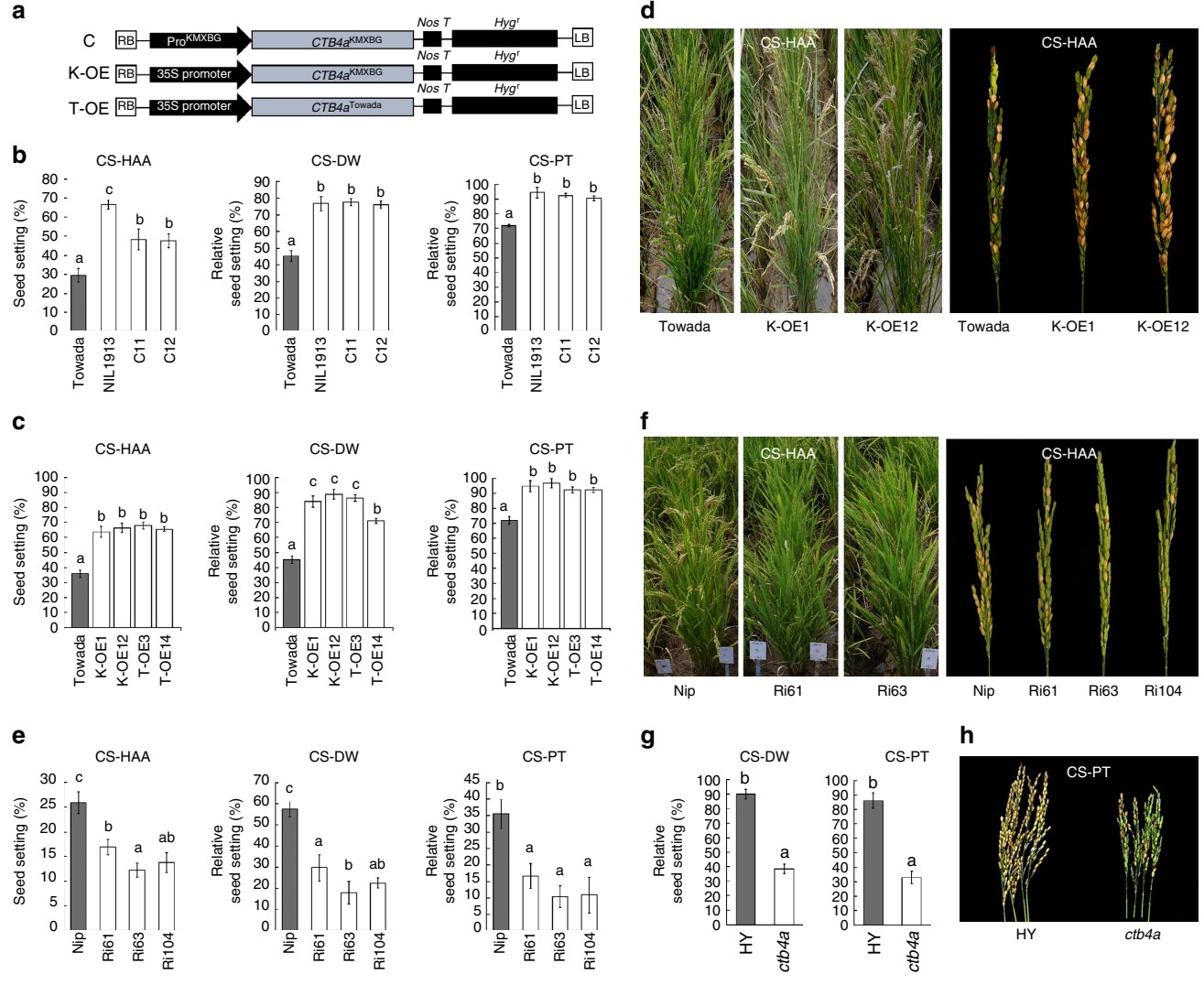

**Figure 2 | Functional analysis of *CTB4a* at the booting stage.** (**a**) Schematic of vectors for transgenic analysis. C, complementation vector; K-OE, *CTB4a*[KMXBG] overexpression vector; T-OE, *CTB4a*[Towada] overexpression vector. (**b**) Statistical results for seed setting of Towada, NIL1913, C11 and C12 under cold stress. Data represent means ± s.d. (*n* = 20). (**c**) Statistical results for seed setting of Towada and overexpression lines under cold stress. Data represent mean ± s.d. (*n* = 25). (**d**) Phenotype of plants and panicles of overexpression lines grown under CS-HAA. (**e**) Statistical results for seed setting of Nip and RNAi lines under cold stress. Data represent means ± s.d. (*n* = 15). (**f**) Phenotype of plants and panicles of RNAi lines grown under CS-HAA. (**g**) Phenotypic analysis of HY and *ctb4a* grown under CS-PT and CS-DW. Data represent means ± s.d. (*n* = 10). (**h**) Panicles of HY and *ctb4a* mutant under CS-PT. The presence of the same lowercase letter above the error bar denotes a non-significant difference between the means (*P* > 0.05, Student's *t*-test).

stable and obviously enhanced cold tolerance compared to Towada under CS-DW, CS-PT and CS-HAA conditions over several years (Fig. 2c,d and Supplementary Figs 6b,8). Similarly, T-OE3 and T-OE14 exhibited enhanced cold tolerance with significantly increased seed setting and showed little difference with K-OE1 and K-OE12 (Fig. 2c and Supplementary Fig. 8). Moreover, we assayed the transcript abundance of *CTB4a* in different overexpression lines and calculated the correlation between *CTB4a* expression level and seed setting. We found that the cold tolerance of different lines had a high correlation ($R^2 = 0.7477$) with *CTB4a* expression level (Supplementary Fig. 9). These results indicate that the Towada allele of *CTB4a* is also functional. The different cold sensitivities of KMXBG and Towada are therefore more likely attributed to nucleotide differences in the promoter region.

To further test the function of *CTB4a* in cold tolerance, an RNA interference (RNAi) vector was constructed and transformed into Nip (as we failed to obtain regenerated plants from calli of KMXBG and NIL1913). Downregulation of *CTB4a* in Nip resulted in a significant reduction in seed setting in RNAi lines under cold stress (Fig. 2e,f and Supplementary Fig. 6c). In addition, a loss-of-function mutant, *ctb4a*, showed significantly reduced seed setting compared to its wild type, Hwayoung (HY), under cold stress (Fig. 2g,h and Supplementary Fig. 10a–c). All of these results suggest a dosage effect of *CTB4a* on cold tolerance at the booting stage.

When rice plants were exposed to low temperatures at the booting stage, anther injury often occurred and led to failure of microspore or pollen development[27]. We found that the anthers of Towada were more seriously injured with a higher proportion of distorted pollen chambers showing unusual patterns of adhesion (Supplementary Fig. 11a–c) and with reduced pollen fertility compared to those of NIL1913 (Fig. 1c,d) and K-OE lines (Supplementary Fig. 12) under cold stress as evidenced by tissue

fractionation, although pistil development was unaffected (Supplementary Fig. 11d). On the contrary, the *ctb4a* mutant showed much lower pollen fertility than HY (Supplementary Fig. 10d,e). When emasculated spikelets of cold stressed Towada and NIL1913 plants were hand-pollinated with pollen from non-cold-stressed plants, there was no significant difference in the seed setting (Supplementary Fig. 11e). These results demonstrate that *CTB4a* confers cold tolerance with increased seed setting via its effect on pollen fertility, but with no apparent effect on female fertility.

KMXBG also showed strong cold tolerance at the vegetative growth stage[25,26]. Strong GUS signals were detected in buds and young leaf sheaths of pCTB4a^KMXBG::GUS transgenic plants (Supplementary Fig. 13). This information prompted us to further investigate the chilling or cold tolerance at the seedling and germination stages. Survival rates of the overexpression and complementation lines were significantly higher than Towada after chilling stress (Supplementary Fig. 14a,b). On the contrary, the RNAi lines showed decreased chilling tolerance compared to Nip (Supplementary Fig. 14c,d). Moreover, the overexpression lines showed much quicker germination than Towada under cold stress (Supplementary Fig. 15). These results indicate that *CTB4a* plays a significant role during vegetative growth as well as at the booting stage.

**Expression pattern of *CTB4a*.** In order to explore the subcellular localization of CTB4a, we introduced a CTB4a-GFP fusion gene under control of the CaMV35S promoter into Nip. Fluorescent signals of GFP were microscopically detected on the cell surface and in chloroplasts in rice leaf sheath cells (Supplementary Fig. 16a). Moreover, the same signal location of GFP and the cell membrane stain dye DiI were observed in root cells (Supplementary Fig. 16b). In addition, transient expression of CTB4a-GFP in tobacco leaf cells confirmed the subcellular localization of CTB4a-GFP to the cell surface and chloroplasts (Supplementary Fig. 16c,d).

To further examine the expression of *CTB4a* in different rice tissues, several experiments were conducted. Quantitative RT-PCR analysis showed that *CTB4a* is widely expressed in root, stem, leaf, sheath and panicle (Supplementary Fig. 17a). Histochemical analysis of pCTB4a^KMXBG::GUS transgenic plants revealed strong GUS activity in different tissues, especially the anthers (Supplementary Fig. 17b). *In situ* hybridization indicated that *CTB4a* was strongly expressed in the tapetum and anther connective vascular bundle (Supplementary Fig. 17c), which was consistent with the effect of *CTB4a* on pollen fertility. This expression pattern of *CTB4a* is consistent with the observed phenotypes in anther tissue.

**CTB4a promoter region variation and cold tolerance.** To test for an association between *CTB4a* alleles and cold tolerance at the booting stage in diverse rice germplasm, we evaluated the cold tolerance of 46 *japonica* and 73 *indica* cultivars (Supplementary Data 1). As expected, *japonica* cultivars showed more cold tolerance than *indica* cultivars (Supplementary Fig. 18). According to the nucleotide polymorphisms identified in the two parents (KMXBG and Towada), we could divide the sequences of the 119 cultivars into nine haplotypes, among them Hap1, Hap7, Hap8 and Hap9 present only in the *japonica* (sub-J), Hap3, Hap4 and Hap6 only in *indica* (sub-I), and Hap2 and Hap5 both in sub-J and sub-I subpopulations. Both parent haplotypes Hap1 (Hap-KMXBG) and Hap9 (Hap-Towada) were present in sub-J (Fig. 3a). Furthermore, phylogenetic analysis showed that the nine haplotypes could be divided into three groups with Hap1 in Group A, Hap2 to Hap7 in Group B, and Hap8 and Hap 9 in

Group C (Fig. 3b). Accessions in Group A (only Hap1) showed obviously stronger cold tolerance than Group B and C (Fig. 3c). Also, 18 wild rice accessions were sequenced and although we found similar diversity to that in cultivated rice accessions, no haplotypes identical to Hap-KMXBG or Hap-Towada were found (Fig. 3a). These results suggest that the cold tolerant haplotype of *CTB4a* may have originated in *japonica* and may be present only in *japonica* (although it is possible that it may be present in accessions not studied here).

All accessions in Hap-KMXBG with five consensus SNPs (SNP-2536, SNP-2511, SNP-1930, SNP-780 and SNP2063) exhibited apparently higher cold tolerance than other haplotypes (Fig. 3a–c). SNP-2536 and SNP-2511 carried *cis* element (gibberellin response element[28] and anaerobic induced regulatory element[29]) changes according to PLACE (Plant *cis*-acting regulatory DNA elements) analysis. We conducted transient assays of the site-directed mutated promoter fragments of *CTB4a* in *Arabidopsis* protoplasts to test the effects of the four SNPs in the promoter region (Fig. 3d). The relative activities of the promoter fragments with SNP-2536, SNP-2511 and SNP-1930 were significantly reduced (Fig. 3d). These results indicate that Hap-KMXBG (Hap1) containing all five consensus SNPs in the promoter region represents the cold-tolerant haplotype and that SNP-2536, SNP-2511 and SNP-1930 are functional SNPs (FNPs) accounting for the expression level differences between the *CTB4a* alleles.

**Low-temperature acclimation of *Tej*-Hap-KMXBG.** In regard to geographic distribution of the nine haplotypes, we found that Hap-KMXBG (Hap1) mostly occurred in northeastern China (N: 40–50°; Supplementary Fig. 19). To examine the detailed geographic distribution of the alleles, we classified the 119 accessions as two kinds of haplotypes, namely Hap-KMXBG and Hap-others. Twenty seven cultivars with Hap-KMXBG were from the northern areas of China, Japan, Korea and Western Europe or from the high-altitude southwestern zone of China (Yunnan-Guizhou plateau with an average elevation of 2,000 m). Hap-others were mainly distributed in regions with higher temperatures, such as southern China and southeastern Asia (Fig. 4a). Moreover, analysis of phylogenetic relationship of *CTB4a* based on 196 SNPs in 459 accessions including 11 wild rice (*O. rufipogon*) lines and 448 varieties from the world rice core collection (Supplementary Data 1) showed that *CTB4a* was quite diverse across the range of *indica*, *japonica*, *Aro* and *Aus* accessions (Fig. 4b). We identified 25 *CTB4a* haplotypes among 459 accessions, and only three haplotypes contained the same FNPs as Hap-KMXBG, but none was wild rice (Fig. 4c). These results imply that Hap-KMXBG does not directly originate from *O. rufipogon* during *japonica* domestication.

The three haplotypes containing the same FNPs as Hap-KMXBG specifically existed in the temperate *japonica* (*Tej*) group suggesting that this unique haplotype, named as *Tej*-Hap-KMXBG, was selected during evolution. We analysed the nucleotide diversity of *CTB4a* among *indica*, *Tej*, tropic *japonica* (*Trj*) and *O. rufipogon* to determine whether selection had acted on *CTB4a*. On average, the nucleotide diversity of *CTB4a* in *Tej* was higher ($\pi = 0.00351$) than in *indica* ($\pi = 0.00223$) or *Trj* ($\pi = 0.00271$). When *Tej* was further divided into *Tej*-Hap-KMXBG and *Tej*-Hap-Towada, the $\pi$ value in *Tej*-Hap-KMXBG ($\pi = 0.00018$) was much lower than that in *Tej*-Hap-Towada ($\pi = 0.00266$), *indica*, *Trj* and *O. rufipogon* ($\pi = 0.00413$; Fig. 4d, Supplementary Fig. 20 and Supplementary Table 2). Such decreased diversity could be a result of positive selection or other demographic situation, such as a bottleneck effect[30,31]. In an attempt to discriminate the two effects, we calculated the

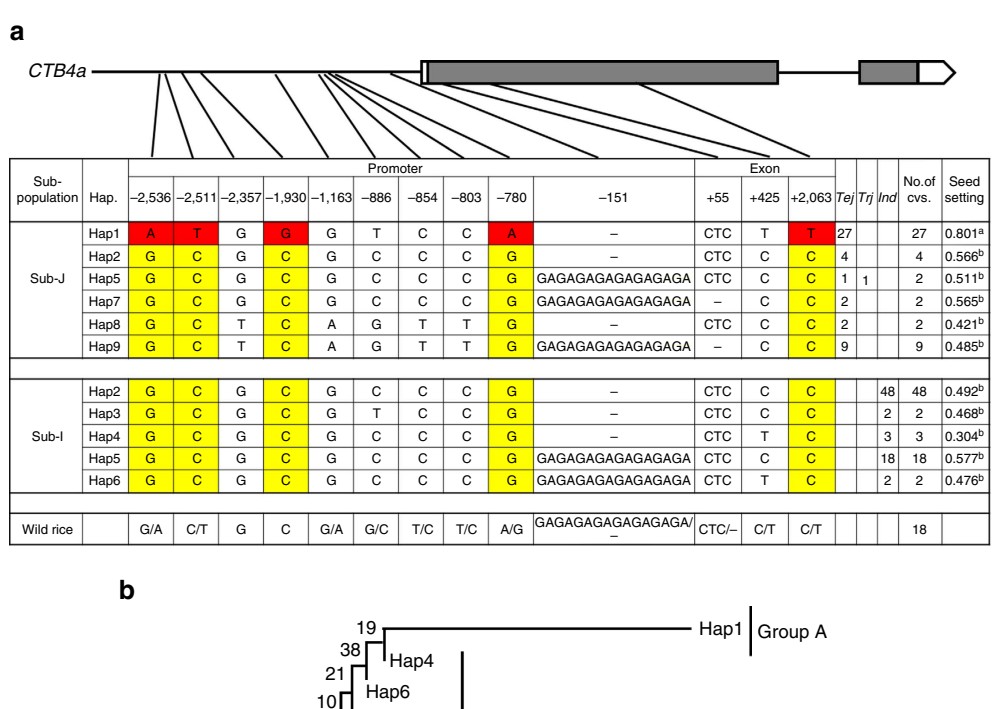

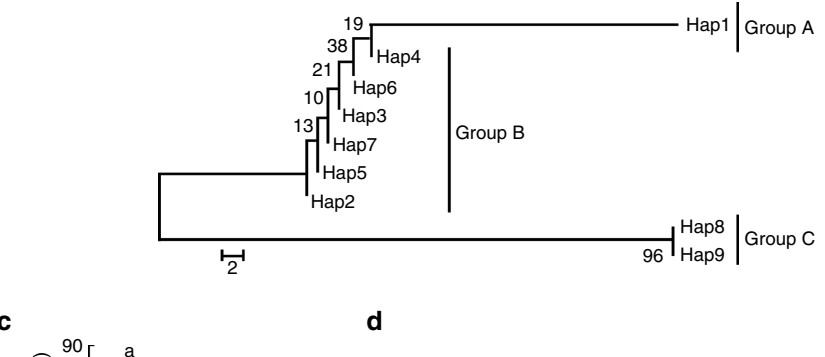

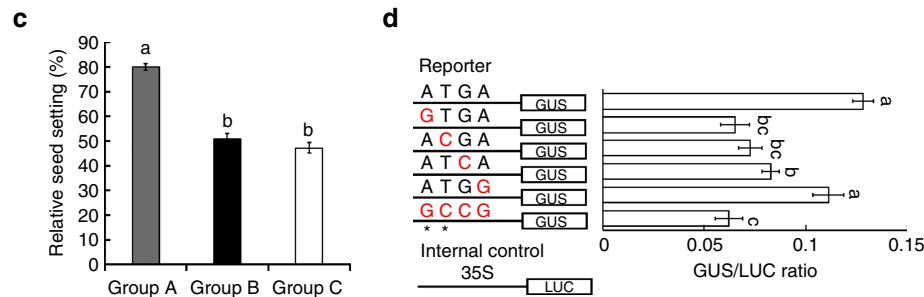

**Figure 3 | Haplotype analysis of CTB4a.** (**a**) Haplotype analysis of the *CTB4a* gene region from 119 rice cultivars and 18 *O. rufipogon* accessions. *Ind*, *indica* population; Sub-J, *japonica* population; Sub-I, *indica* population; *Tej*, temperate *japonica* population; *Trj*, tropical *japonica* population. Data represent means (*n* = 5). (**b**) Phylogenetic tree of the nine haplotypes divided into Groups A (Hap1), B (Hap2- Hap7) and C (Hap8- Hap9). The scale bar indicates average number of substitutions per site for different haplotypes. (**c**) Relative seed setting of the three Groups. Data represent means ± s.d. (**d**) Transient expression assay of promoter activity in *Arabidopsis* protoplasts. Left, constructs with site-directed mutations at the four SNPs in the promoter region. Two *cis*-elements changed at SNP2536 and SNP2511 indicated with '*'. Right, relative GUS/LUC values. The presence of the same lowercase letter above the error bar denotes a non-significant difference between the means (*P* > 0.05, Student's *t*-test).

nucleotide diversity of 20 kb flanking regions of *CTB4a*. The average nucleotide diversity of the *CTB4a* flanking regions in *Tej*-Hap-KMXBG was significantly lower than that among others (*P* < 0.001, student's *t*-test; Supplementary Table 3). As positive selection will result in range of genomic changes, resembling to our observation on the polymorphism pattern of *Tej*-Hap-KMXBG, the decrease of nucleotide diversity of *CTB4a* in *Tej* may be largely caused by positive selection. Also significant Tajima's *D* (ref. 32) values were observed in *Tej*-Hap-KMXBG cultivars (Supplementary Table 2), which were consistent with selection at *CTB4a* locus. To further determine whether the reduction of nucleotide diversity in *Tej*-Hap-KMXBG was caused by artificial selection, we used Maximum likelihood Hudson–Kreitman–Aguade (MLHKA) tests on *CTB4a* sequences for six

groups and seven neutral genes[33] as a control. A significant value was observed for *Tej*-Hap-KMXBG (*P* = 3.132E-05, student's *t*-test) indicating that a strong artificial selection occurred on the *CTB4a* locus during *japonica* domestication (Supplementary Table 2). These results collectively indicate that *Tej*-Hap-KMXBG of *CTB4a* representing a new allele generated from natural variation in cooler environment is retained by artificial selection for low-temperature acclimation in rice.

**CTB4a interaction with AtpB.** To explore the potential mechanism by which CTB4a influences cold tolerance, we screened for interacting proteins using truncated CTB4a with the kinase domain (CTB4a^KD) as bait and identified three candidates,

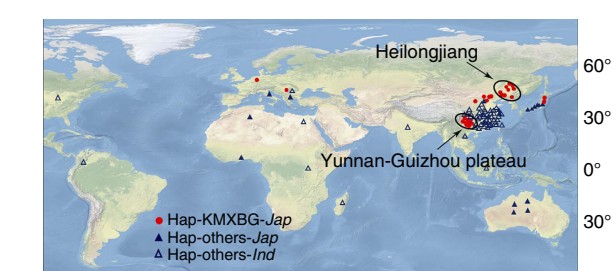

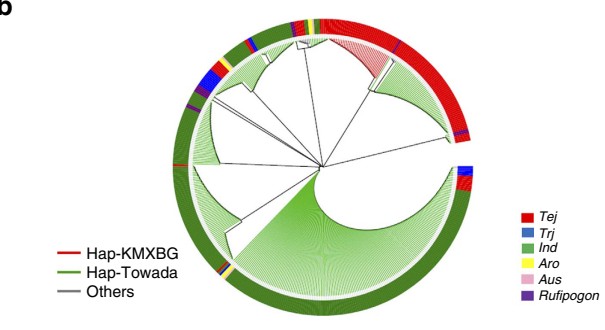

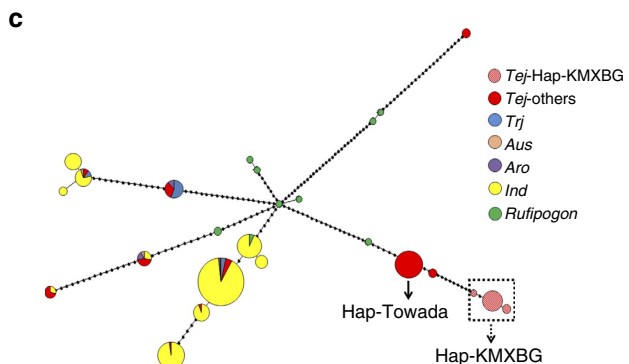

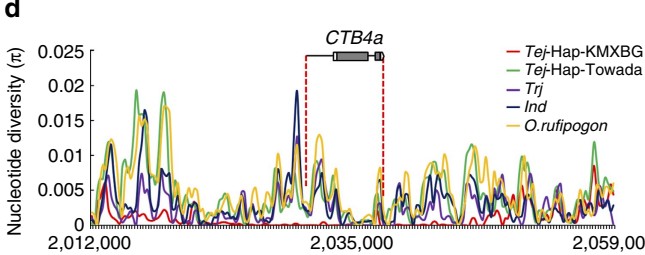

**Figure 4 | The geographic and phylogenetic origins of *CTB4a*.**
(**a**) Geographic distributions among 119 accessions. The *japonica* Hap-KMXBG members are indicated by red circles. Other *japonica* and *indica* accessions are indicated by solid blue or hollow triangles, respectively. (**b**) Phylogram of *CTB4a* generated from 459 diverse rice accessions including the *Tej*, *Trj*, *Ind*, *Aro*, *Aus* and *O. rufipogon* showing divergence between the Hap-KMXBG and Hap-Towada. (**c**) Haplotype network of *CTB4a*. Circle size is proportional to the number of samples for a given haplotype. Black spots represent unobserved, but inferred haplotypes. Lines between haplotypes represent mutational steps between alleles. The black square with dotted arrow encloses rice lines with the *CTB4a* Hap-KMXBG type SNPs and the solid arrow indicates the *CTB4a* SNPs of Hap-Towada type. (**d**) Nucleotide diversity of *CTB4a* in cultivar populations and the wild rice group. The X-axis denotes the position of *CTB4a* and the Y-axis indicates average π values.

two of which were beta subunits of ATP synthases and one of which was named AtpB (Fig. 5a and Supplementary Table 4). Interaction between CTB4a$^{KD}$ and AtpB *in vitro* was confirmed by a GST pull down assay (Fig. 5b) and *in vivo* by a co-immunoprecipitation (Fig. 5c). Bimolecular fluorescence complementation assays performed in *Nicotiana benthamiana* showed that both the CTB4a$^{KMXBG}$-YFP$^N$ and CTB4a$^{Towada}$-YFP$^N$ could interact with AtpB-YFP$^C$ at chloroplasts, but not YFP$^C$ alone (Fig. 5d). Next, we found that ATP synthase activity and ATP content as well as photosynthetic rate, were significantly higher in leaves of NIL1913 and the overexpression lines than in Towada (Fig. 6a), but were significantly lower in *ctb4a* compared to HY at the booting stage under cold stress (Fig. 6b). Also higher ATP content were observed in panicles of NIL1913 and the overexpression lines than in Towada (Fig. 6c), and reduced ATP content in panicles of *ctb4a* compared to HY under cold stress (Fig. 6d). Moreover, exogenous application of ATP partly rescued the cold sensitivity of Towada (Fig. 6e). ATP supplies energy for plant growth and metabolism. We found that NIL1913 and overexpression lines showed higher yield with increased filled grain number per-panicle and thousand-kernel weight than Towada under cold stress (Fig. 6f and Supplementary Fig. 21).

To further investigate the function of *AtpB* for cold tolerance, we obtained *AtpB* overexpression lines and found that *AtpB*-OE3 and *AtpB*-OE24 showed significantly increased seed setting than that of Nip under cold stress at the booting stage (Fig. 6g and Supplementary Fig. 22). These results collectively indicate that CTB4a can interact with AtpB. It is possible that this interaction mediates ATP synthesis to ensure an adequate energy supply to enhance seed setting and increase yield in cold rice growing areas.

## Discussion

In general, seed setting of plants stressed by natural low temperatures or artificially controlled low temperatures is used to evaluate the cold tolerance at the booting stage in rice[34]. However, the differences in rice growth period and the limitations of equipment used for cold evaluation have restricted the accuracy which with cold tolerance at the booting stage in rice can be assessed. Compared with other agronomic traits, the evaluation of cold tolerance of rice, especially at the booting stage, is more difficult. For example, a total of 16 years were spent for fine mapping and cloning of *Ctb1* (ref. 24). Based on these reasons, we constructed a series of cold-tolerant near-isogenic lines using KMXBG, a strong cold-tolerant native Yunnan cultivar, as the donor parent and Towada as the recipient parent to exclude the influence of the different heading date and genetic backgrounds, which laid the foundation for the isolation of genes for cold tolerance at the booting stage. In addition, we established three methods to evaluate the cold tolerance of the plant materials. High-elevation environments where low-temperature stress occurs provides a reliable location to study a large number of genotypes, which is necessary for the application of map-based cloning of the QTLs for cold tolerance.

Low temperatures lead to a series of changes at the molecular and cellular levels as well as physiological and biochemical levels in plant cells. Plants need energy to reinforce resistance to cold stress, among which ATP is one of the important energy sources[35–38]. Our results demonstrated that upregulation of *CTB4a* could increase ATP synthase activity and ATP content in NIL1913 and overexpression lines under cold stress at the booting stage (Fig. 6a–d). Higher pollen fertility was observed in NIL1913 and *CTB4a* overexpression lines than Towada under cold stress (Fig. 1c,d and Supplementary Fig. 12). Moreover, the application of exogenous ATP increased seed setting of Towada under cold stress (Fig. 6e). We therefore suggest that the low expression of

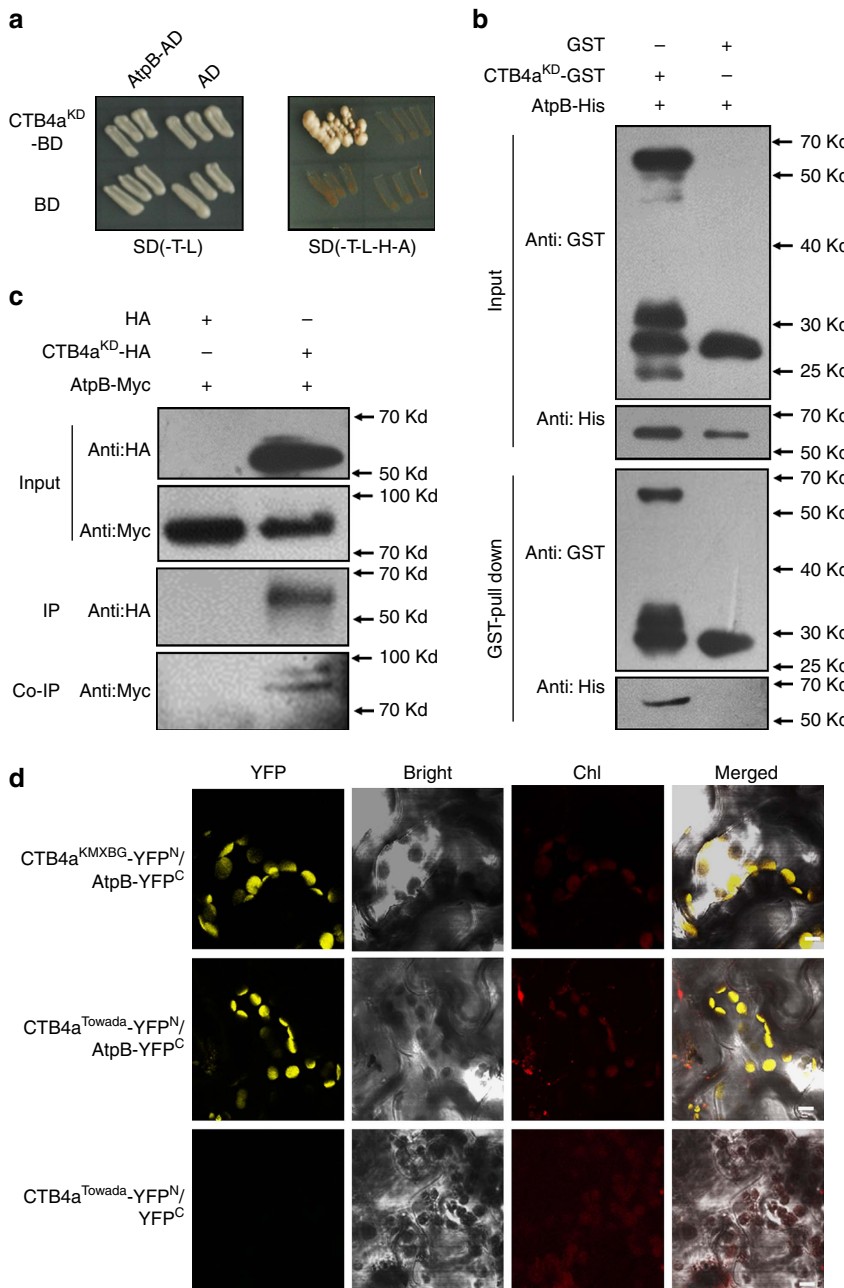

**Figure 5 | CTB4a can interact with AtpB.** (**a**) Interaction assay of CTB4a[KD] and AtpB in yeast (strain AH109). Interaction was determined by a growth assay on medium lacking Trp-Leu-His-Ade. (**b**) GST-pull down assay confirming the interaction between CTB4a[KD] and AtpB *in vitro*. Purified CTB4a[KD]-GST and AtpB-His fusion protein from *E.coli* BL21 were pull-downed by GST beads and blots were probed with anti-GST or anti-His. (**c**) Interaction assay of CTB4a[KD] and AtpB *in vivo*. Co-expressed CTB4a[KD]-HA or HA and AtpB-Myc in tobacco leaves were immunoprecipitated by anti-HA antibody and blots were probed with anti-Myc or anti-HA. (**d**) Bimolecular fluorescence complementation assay. Chl, chloroplast; YFP, yellow fluorescent protein. Scale bar, 5 μm.

*CTB4a* and reduced ATP content in Towada lead to less energy for cold tolerance and reduced pollen fertility and decreased seed setting. It was previously reported that a shortage of ATP can reduce grain productivity[39,40]. We finally found that NIL1913 and *CTB4a* overexpression lines exhibited improved grain yield under CS-HAA (Fig. 6f).

Phosphorylation can affect ATP synthase activity or assembly, and this modification often occurs in the β subunit which plays catalytic function[41]. The β subunit of chloroplast ATP synthase can be phosphorylated by kinases and this has been reported in a variety of plants[42,43]. However, the phosphorylation of ATP synthase has not been well documented in rice. We found that

CTB4a exhibited autophosphorylation activity but could not phosphorylate AtpB *in vitro* (Supplementary Fig. 23). Many plant RLKs undergo a signal perception, self-phosphorylation, dimerization and *trans*-phosphorylation processes to activate basal kinase function[44]. Our data indicated that CTB4a can interact with AtpB, and that upregulation of *CTB4a* increased ATP synthesis. Overexpression of *AtpB* also increased cold tolerance of rice at the booting stage (Figs 5 and 6a,g). It is possible that AtpB may be phosphorylated by CTB4a *in vivo*. In addition, it could also be possible that CTB4a interacts with AtpB to enhance the phosphorylation of AtpB by other kinases, thus affecting ATP synthase activity or may regulate ATP production

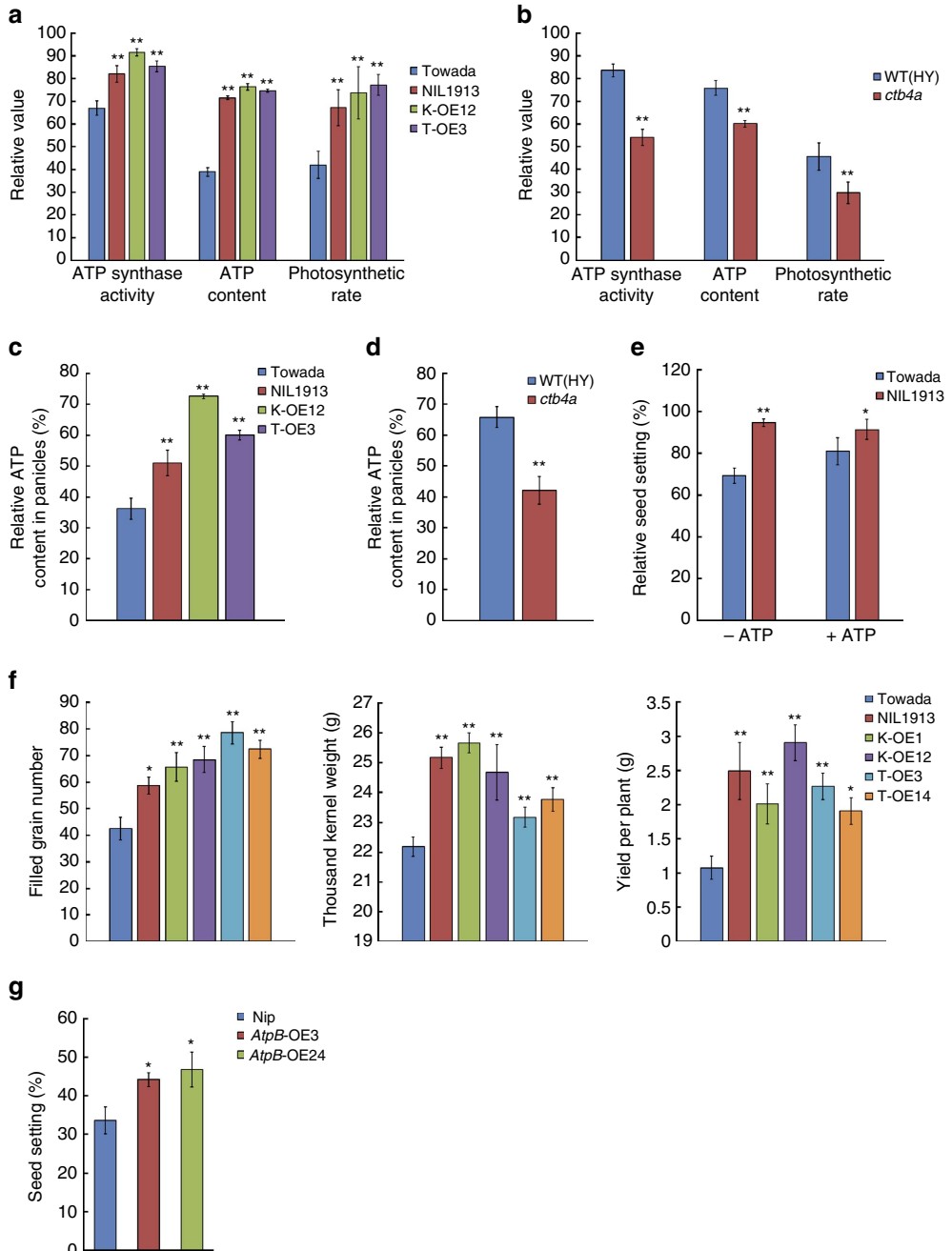

**Figure 6 | *CTB4a* mediates ATP supply and improves rice yield.** (**a**) ATP synthase activity, ATP content and photosynthetic rate in leaves of Towada, NIL1913 and overexpression lines under CS-PT for 3 days. (**b**) ATP synthase activity, ATP content and photosynthetic rate in leaves of HY and *ctb4a* mutant under CS-PT for 3 days. (**c,d**) ATP content in panicles of overexpression lines (**c**) and *ctb4a* mutant (**d**). (**e**) Relative seed setting of Towada and NIL1913. Exogenously applied ATP (5 mM) rescued the cold sensitivity of Towada. Data represent means ± s.d. (n = 15). (**f**) Filled grain number per-panicle, thousand-kernel weight and yield per-plant in Towada, NIL1913 and transgenic plants under CS-HAA. Data represent means ± s.d. (n = 30). (**g**) Statistical results for seed setting of Nip and *AtpB* overexpression lines under CS-PT. Data represent means ± s.d. (n = 5). All P values were calculated by Student's t-test. **P < 0.01, *P < 0.05.

in a different manner, such as via the assembly of the ATP synthase complex under cold stress conditions. These hypotheses remain to be verified in the further studies.

The progress in cloning *CTB4a* began with a tightly linked QTL in the gene region (Fig. 1e and Supplementary Fig. 3a). The mining and pyramiding of important QTLs is an effective method to produce varieties with high yield, good quality and stress tolerance. CTB4a is a conserved LRR-RLK in plants (Supplementary Fig. 4b), especially in gramineous plant where protein identity exceeded more than 66% (Supplementary

Fig. 4c). Use of *CTB4a* and its homologous genes could potentially accelerate production and use of cold tolerant varieties in crop plants. During crop domestication, the genetic modification of a wild species could alter phenotypic characteristics to meet human demands, in which natural selection plays an important role to maintain the genetic diversity of cultivars[45,46]. We found that *Tej*-Hap-KMXBG of *CTB4a* had been retained in landraces and recently improved varieties, such as DAOHUAXIANG, KENDAO 12, DONGNONG 428, LIJIANGXINTUANHEIGU, YUNJING 23 and YUNDAO 1, of

temperate *japonica* in northeastern China and the Plateau of Lijiang (altitude 2, 384 m), Yunnan (Supplementary Data 1). With wide spread planting of varieties with *Tej*-Hap-KMXBG of *CTB4a*, Heilongjiang province (N:43–53°), referred to as the 'cold rice growing zone' (Supplementary Fig. 24), has become the largest rice area in China at 4.3 million hectares per year. It is expected that *Tej*-Hap-KMXBG of *CTB4a* will allow rice production to expand in the currently cooler areas.

To our knowledge, *CTB4a* is the first LRR-RLK gene conferring cold tolerance at the booting stage isolated by forward genetics in rice. More importantly, NIL1913 and *CTB4a* overexpression lines showed stable cold tolerance not only in the greenhouse but also in field conditions over several years (Fig. 2 and Supplementary Figs 2,8). On the basis of the alleles and FNPs identified in this work, we suggest that *CTB4a* has great potential for improving rice cold tolerance at the booting stage via molecular breeding techniques. New cultivars designed on the basis of our study could test whether *CTB4a* can confer cold tolerance in other crops in the future.

In summary, this work has uncovered a novel gene *CTB4a* controlling cold tolerance at the booting stage in rice. Our findings suggest that the natural variation in the promoter region of *CTB4a* may have enhanced rice adaptation to cold habitats during domestication. We propose that *CTB4a* confers resistance to cold stress by mediating ATP supply to reinforce pollen fertility at low temperature. This gene discovery could be of potential significance for understanding crop domestication and future molecular breeding in crops.

## Methods

**Plant materials.** In previous study, we developed a set of cold-tolerant NILs by backcrossing a cold-tolerant landrace, KMXBG (*O. sativa* ssp. *japonica* cv), as donor to a cold-sensitive variety, Towada (*O. sativa* ssp. *japonica* cv). NIL1913 containing QTL *qCTB4-1* was constructed by six-times backcrossing the KMXBG × Towada hybrid to Towada to generate a $BC_6F_2$ population. Towada and Nip were used for transformation. $CTB4a^{KMXBG}$ overexpression transgenic lines, $CTB4a^{Towada}$ overexpression transgenic lines and complementation lines in Towada background, as well as RNAi lines, AtpB overexpression lines in Nip background were used to evaluate cold tolerance. The *ctb4a*, T-DNA insertion mutant in HY (*O. sativa* ssp. *japonica* cv.) background, was purchased from the Korean Rice Mutant Center[47].

**Evaluation of cold tolerance.** Three methods were adopted to evaluate cold tolerance at the booting stage. The mapping population was planted under CS-HAA conditions in the summers of 2007, 2008 and 2009. About 30-day-old seedlings were transplanted ($L \times W = 12.5\,cm \times 25\,cm$) in the field. Cold tolerance of the $F_2$ population was measured by spikelet fertility of the main panicles at seed ripeness, and $F_3$ families were evaluated as average spikelet fertilities of the main panicles of 15 plants per line. $F_4$ families from homozygous recombinant individuals were evaluated as average spikelet fertilities of the main panicles from 10 plants in each line.

The phenotypes of transgenic lines and control plants were evaluated under CS-DW and CS-PT in Beijing, and CS-HAA in Kunming as mentioned above. For CS-DW, the main panicles at booting were labelled and then transferred to deep cold water (16–18 °C). After one week, the plants were transplanted into the field. The spikelet fertilities of labelled panicles were investigated for evaluating cold tolerance. A similar method to CS-DW was adopted for CS-PT, except that the plants were grown in a phytotron (16–17 °C).

For evaluating cold tolerance at the seedling stage, surface-sterilized seeds were first soaked in water at 28–30 °C for 2 days and germinated seedlings with similar vigour were planted in pots containing a mixture of soil and vermiculite, and grown in a growth chamber supplied with Hoagland's solution. Four-week-old seedlings were treated with low temperature (4–5 °C) for 7 days and then returned to normal conditions for 10 days. Phenotypes were evaluated by survival rates. At least five pots of 20 plants were assessed for each line.

Germination tests were conducted as described by Fujino *at al.*[19] with minor changes. After breaking dormancy, 50 seeds were placed on a filter paper in a 9 cm Petri dish covered with distilled water. The dishes were placed in an incubator at 16 °C. The number of the germinated seedlings was recorded.

**Map-based cloning of *CTB4a*.** High-resolution mapping was conducted using $BC_6F_2$ populations derived from the cross between NIL1913 and Towada with Towada as the recurrent parent. Two $F_2$ populations were used for fine mapping.

One $F_2$ population consisting of 179 individuals was grown at the experimental farm, Yunnan Academy of Agricultural Sciences, Kunming, in the summer of 2007. Another $F_2$ population containing 3,102 individuals derived from the same cross was grown at the China Agricultural University Experiment Station at Sanya (19° N, 109° E), Hainan province, in the winter of 2007. All $F_3$ families come from the first $F_2$ population, and $F_3$ families from 46 recombinant individuals identified in the second $F_2$ population were grown at Kunming in the summer of 2008 for evaluation of cold tolerance. Gene annotation was conducted in RGAP (http://rice.plantbiology.msu.edu/). The primer sequences for genotyping are provided in Supplementary Data 2.

**Vector construction and genetic transformation.** For construction of a complementation plasmid, a 7.8 Kb DNA fragment containing the 3.2 kb promoter, the full 4.1 kb exons and intron region and a 0.5 Kb 3′-UTR of *CTB4a* was amplified from KMXBG, digested with *Pme*I and *Asc*I, and cloned into binary vector pMDC83 (ref. 48). For construction of an overexpression plasmid, the full coding sequences were amplified from the cDNA of KMXBG and Towada, digested with *Asc*I and *Pac*I, and cloned into binary vector pMDC32 (ref. 48), respectively. To construct an RNAi plasmid, the unique fragment was amplified from Nip, digested with *Sac*I and *Spe*I, and cloned into pTCK303 vector[49] to generate the forward insertion. Further dsRNAi fragments obtained by digestion with *Bam*HI and *Kpn*I were cloned into the same vector to generate the reverse insertion. For construction of GUS plasmid, 3 kb DNA fragments containing the *CTB4a* promoters from KMXBG and Towada were amplified, digested with *Pme*I and *Asc*I, and inserted into the pMDC162 vector[48], respectively. For construction of the GFP plasmid, the ORF of *CTB4a* without stop codon was amplified from the cDNA of KMXBG, digested with *Spe*I and *Asc*I, and inserted into the pMDC83 vector[48]. All fragments were amplified by the high fidelity PCR enzyme KOD-FX (TOYOBO, KFX-101). Primer sequences for vector constructions are provided in Supplementary Data 2.

All plasmids confirmed by sequence were introduced into *Agrobacterium tumefaciens* strain EHA105 and transferred into recipient materials by the *Agrobacterium*-mediated method[50].

**Expression pattern analysis.** For cold-induced expression analysis of *CTB4a* at the booting stage, NIL1913 and Towada were stressed under CS-PT, and the panicles and leaves were sampled at different time points for RNA extraction. For cold-induced expression analysis of *CTB4a* at the seedlings stage, leaves of NIL1913 and Towada were sampled from 4–5 °C conditions at different time points for RNA extraction. For tissue expression pattern analysis, different tissues of KMXBG and Towada were sampled for RNA extraction and quantitative RT-PCR. Total RNA was extracted from different plant tissues using RNAiso Plus (Takara, D9108B). Each experiment was performed with three biological samples and each with three technical replications. *OsActin1* was used as a reference. The PCR primer sequences are given in Supplementary Data 2. $T_3$ homozygous transgenic plants containing the $pCTB4a^{KMXBG}$::GUS vector were used for GUS histochemical staining.

**Subcellular localization.** Leaf sheaths of CaMV35S::GFP and CaMV35S::CTB4a-GFP transgenic plants were used to determine the subcellular location. Root cells of CaMV35S::CTB4a-GFP transgenic plants stained with DiI were used for detection of fluorescence signals on the plasma membrane. CaMV35S:CTB4a-GFP vector and plasma membrane marker CD3-1007 (ref. 51) were transformed into *Agrobacterium tumefaciens* strain EHA105 and co-infiltrated into leaves of *N. benthamiana* with the suspension containing 10 mM 2-(N-morpholino) ethanesulfonic acid, 10 mM $MgCl_2$ and 150 μM acetosyringone together with p19 (ref. 52) as previously described. After 3 days incubation at 25 °C, the tobacco leaves were used for fluorescence signal observation.

Green and red fluorescence were observed under a confocal microscope (OlympusFV1000). GFP was excited with a 488 nm laser, CD3-1007 and DiI were excited with a 543 nm laser. The emission spectra were collected at 500–550 nm for GFP, and 565–615 nm for CD3-1007 and DiI. To detect auto-fluorescence of chlorophyll, samples were examined with a long-pass 630 nm filter set.

**Microscopy.** For the morphological observations of anther, samples were fixed in FAA solution and then dehydrated through a graded ethanol and embedded in paraffin. Then 8–10 μm thickness sections were obtained and stained using 0.025% toluidine blue for 30 min, after which the washed sections were observed microscopically for photograph (Olympus SEX16). Pistils fixed with FAA solution were observed directly. For measuring pollen fertility, fixed anthers were ground to release pollen grains and then stained with $I_2$–KI solution. Blue stained pollen grains were counted to determine pollen fertility.

For *in situ* hybridization experiment, the spikelets at about stage10[53] were collected and fixed in 4% (w/v) paraformaldehyde at 4 °C overnight, followed by a series of dehydration by graded ethanol infiltration and finally were embedded in paraffin. Then 8 μm thickness sections were obtained using microtome (Leica). 131 bp of *CTB4a* cDNA was sub-cloned into pMD19-T vector and used as template to generate antisense and sense probes. The digoxingenin labelled probes were prepared using digoxingenin RNA Labeling Kit (Roche, 1175025). After hybridization, the detection of digoxingenin-labelled was conducted using

digoxingenin Nucleic Acid Detection Kit (Roche, 11175041910). The primer sequences used for probe preparation are given in Supplementary Data 2.

**Haplotype analysis and SNP identification.** DNA sequences of *CTB4a* in 137 rice accessions were amplified by PCR using primers listed in Supplementary Data 2. The PCR products were sequenced and used to perform haplotype analysis. Differences in phenotypic values and haplotypes were examined by one-way ANOVA or Student's *t*-tests. The Duncan multiple range test was conducted to make further comparisons when the results of the analyses were significant ($P < 0.05$). For one population sequence set a 47 kb segment, centred on *CTB4a*, was obtained from the rice 3 K project[54] (RFGB, http://www.rmbreeding.cn/Index/s), with a missing rate of $\leq 80\%$. A total of 295 *indica*, 144 *japonica*, 11 *O. Rufipogon*, 6 *Aro* and 3 *Aus* accessions were retained for analysis. In addition, SNPs in the 47 kb region were used for the variety phylogenetic comparison of the *CTB4a* region.

**Phylogenetic analysis of *CTB4a*.** A neighbor-joining variety tree (1,000 replications of bootstrap tests) of rice varieties was constructed using MEGA 5.0 (ref. 55). The resulting tree was visualized and annotated using EvolView[56]. A total of 196 SNPs were used to analyse in the 7 kb region containing *CTB4a*. A panel of 448 rice and 11 *O. Rufipogon* accessions was used to construct a minimum spanning tree for *CTB4a*. Arlequin version 3.5 (ref. 57) was used to define the haplotypes and to calculate the minimum spanning tree among haplotypes. Arlequin's distance matrix output was used in Hapstar-0.6 (ref. 58) to draw a minimum spanning tree.

**Evaluation of artificial selection.** The effects of artificial selection were evaluated according to the π value of the 47 kb sequence. Nucleotide diversity was calculated by a custom PERL script. Different groups of rice accessions were examined including *Tej*, *Tej*-Hap-KMXBG, *Tej*-Hap-Towada, *Trj*, *Ind* and *O. Rufipogon* accessions. Statistical differences in averaged nucleotide diversity among the upstream 20 kb region, the middle 7 kb region including *CTB4a* and downstream 20 kb region were assessed for each subpopulation. Tajima' *D* test of *CTB4a* in different groups was calculated and performed using DnaSP 5.10 (ref. 59). MLHKA[60] tests were used to detect the departure of *CTB4a* polymorphisms from neutrality with a set of known neutral genes, including *Adh1*, *GBSSII*, *Ks1*, *Lhs1*, *Os0053*, *SII1* and *TFIIAγ-1* (ref. 33).

**Transient expression in *Arabidopsis* protoplasts.** For promoter activity analysis, a series of mutated pCTB4a::GUS fusion constructs were used for transient transformation into *Arabidopsis* protoplasts. The CaMV35S::LUC plasmid was used as an internal transformation control to provide an estimate of the extent of transient expression. The co-transferred protoplasts were incubated for 16 h at 25 °C and 3 h at 4 °C before harvest. Sampled protoplasts were lysed in 100 μl lysis buffer (Promega, E4550). After centrifugation, the supernatant was used for glucuronidase (GUS) and firefly luciferase (Luc) activity analysis. GUS activity was assayed with 1 mM 4-methylumberlliferyl-β-d-glucuronide in lysis buffer. The reaction was terminated with 0.2 M Na$_2$CO$_3$ after 30 min, the reaction product MU was measured using Fluorescence FLx800 microplate fluorescence reader (BIO-TEK Instruments). LUC activity was also determined by Fluorescence FLx800 microplate fluorescence reader. Ratios of GUS to LUC activities were used to define relative promoter activity. Three biological replicates, each with five technical replicates, were assayed for each construct. The *cis*-element analysis was conducted in PLACE (http://www.dna.affrc.go.jp/PLACE).

**Yeast two-hybrid assay.** In order to determine proteins interacting with CTB4a in rice, truncated cDNA of CTB4a containing a kinase domain (CTB4a$^{KD}$) was amplified and sub-cloned into the pGBKT7 vector. CTB4a$^{KD}$ protein without activation activity was used as bait to screen a cDNA library prepared from equal amounts of poly-(A) containing RNA from leaves and panicles after three days of cold stress at 16 °C. Experimental procedures for screening and plasmid isolation were performed according to the manufacturer's user guide (Clontech, PT3024-1). Yeast strain AH109 was used in this assay. Primer sequences are provided in Supplementary Data 2.

**In vitro GST-pull down assay.** In order to confirm the interaction between CTB4a$^{KD}$ and AtpB *in vitro*, CTB4a$^{KD}$ and AtpB were amplified and sub-cloned into pGEX-4T-1 and pET28a, respectively. The plasmids were transformed into *E.coli* BL21. Then 5 μg of purified CTB4a$^{KD}$-GST or GST and AtpB-His protein were incubated in 1 ml of PBS buffer (pH 7.4, 0.1% NP-40) at 4 °C with gentle agitation for 2 h before addition of 20 μl of Glutathione Sepharose 4B beads (GE healthcare, cat.no.17-0756-01), and continued incubation for 1 h. The Glutathione Sepharose beads was collected by brief centrifugation, washed five times in PBS buffer, re-suspended in SDS loading buffer, subjected to SDS-PAGE electrophoresis, and probed with an anti-GST (Sigma, SAB5300159, dilution, 1:1,000) and anti-His antibody (Sigma, H1029, dilution, 1:2,000). The original western blot images are provided in Supplementary Fig. 25. Primer sequences are provided in Supplementary Data 2.

**Co-immunoprecipitation assay.** The Super::AtpB-Myc and CaMV35S::HF-CTB4a$^{KD}$ plasmids were individually transformed into *Agrobacterium tumefaciens* strain EHA105. The strains were co-infiltrated into tobacco leaves with p19. Total proteins were extracted from tobacco leaves expressing Super::AtpB-Myc/CaMV35S::HF-CTB4a$^{KD}$ or Super:: AtpB-Myc/CaMV35S::HF using 300 μl of IP buffer (50 mM Tris–HCl, pH 7.5, 150 mM NaCl, 0.1% NP-40, 3 mM DTT and 1 × Protease Inhibitor Cocktail, Roche). Lysates at 4 °C were centrifuged at 12,000 g for 20 min and 40 μl was taken as input. The left supernatants were incubated with 20 μl HA agarose beads (Sigma, A2095) at 4 °C for at least 2 h in a 1 ml medium supplemented with IP buffer. Subsequently, beads were collected by centrifugation at 5,000 g for 3 min, washed five times with IP buffer and eluted with boiled 1 × SDS loading buffer. Samples were then separated in SDS-PAGE gels and detected by anti-Myc (Sigma, M4439, dilution, 1:3,000) and anti-HA antibodies (Sigma, H3663, dilution, 1:3,000). The original western blot images are provided in Supplementary Fig. 25. Primer sequences are provided in Supplementary Data 2.

**Determination of ATPase activity and ATP content.** Panicles and leaves of different materials under normal (28 °C) and CS-PT for 3 days were sampled for determination of ATP synthase activity and ATP content. ATP synthase activity was determined by the protocol provided with the ATPase Activity Assay Kit (GenMed Scientifics, GMS50248.3). ATP was extracted in lysis buffer (50 mM Tris–HCl, pH 7.5, 100 mM NaCl, 1 mM EDTA, 0.2% TritonX-100 and 2% glycerol) and quantified based on the requirement of luciferase for ATP in producing light following the manufacturer's protocol supplied with the ATP Determination Kit (Invitrogen, A22066). A standard curve of ATP concentrations from 0.01 μM to 3 μM was used in the analysis. Each experiment was conducted with three biological samples, each with five technical replicates.

**Measurement of photosynthetic rate.** The photosynthetic rates of different materials under normal and CS-PT conditions were measured using a LICOR-6400 CO$_2$ gas exchange analyser (LICOR-6400, Lincoln, NE). Statistical analysis was based on data obtained from at least eight plants of each line.

**In vitro phosphorylation assay.** The kinase activity assays of CTB4a$^{KD}$ were carried out in a 15 μl reaction mixture containing CTB4a$^{KD}$-GST (2 μg)/GST (2 μg), CTB4a$^{KD}$-GST (2 μg)/AtpB-GST (2 μg) or GST (2 μg)/AtpB-GST. The kinase buffer contained 50 mM Tris–HCl (pH 7.5), 10 mM MgCl$_2$ and 20 mM ATP. The reactions were started by adding 10 μCi of [γ-$^{32}$P] ATP and the samples were incubated for 30 min at 30 °C. The reaction was stopped by adding 3 μl 6 × loading buffer and boiled for 5 min, then the proteins were separated by 10% SDS-polyacrylamide gel electrophoresis and detected by Typhoon 9410 imager (GE Healthcare). Primer sequences are provided in Supplementary Data 2.

**Data availability.** Accession code: the genomic DNA sequence of *CTB4a* in KMXBG has been deposited in GenBank with the accession number FJ693708.1.The authors declare that all data supporting the findings of this study are available within the manuscript and its supplementary files or are available from the corresponding author upon request.

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

## Acknowledgements

We thank Robert A. McIntosh (University of Sydney) and Jose M. Alonso (North Carolina State University) for critical reading and suggested revisions for the manuscript, S. Ge and Lian Zhou (Institute of Botany, Chinese Academy of Sciences) and X.M. Zheng (Institute of crop sciences, Chinese Academy of Agricultural Sciences) for the help with Tajima's D values and MLHKA test analysis, Fengmei Li, Jinli Miao, Xin Wang and Yanfa Chen for preparing the samples, Xueqiang Wang, Yan Zhao and Xiaoyang Zhu for help with data analysis. This work was supported by grants from Ministry of Science and Technology of China (2016YFD0100300, 2015BAD02B01), the National Natural Science Foundation of China (31671649, 31471456), the Ministry of Agriculture of China (2014ZX08001-003-002 and 2014ZX08009-003-002), Chairman of Guangxi Autonomous Region (1517-03) and Key Program of Guangxi Academy of Agricultural Sciences (2016JZ05).

## Author contributions

Z.Z., Jinjie Li and Z.L. designed the research, and together with Y.P., Jilong Li, L.Z. and H.S. performed most of experiments and analysed the data. Y.Z. constructed the NIL line and was responsible for cold tolerance evaluation under CS-HAA. H.G., Shuming Yang, W.Z., J.Y., X.S., G.L., Y.D., L.M., S.S. and L.D. performed part of the experiments. H.Z., Shuhua Yang and Y.G. conceived and supervised the project. Z.Z., Jinjie Li and Z.L. conceived the experiment and wrote the manuscript.

## Additional information

**Competing interests:** The authors declare no competing financial interest.

