## [Peer Review File · Nature Communications]

Reviewers' comments:

Reviewer #1 (Remarks to the Author):

The manuscript titled "Natural Variation in CTB4a Enhances Rice Adaptation to Cold Habitat" by authors, Zhang et al. have explored and cloned a QTL, CTB4a, which conferring cold tolerance at both booting stage and vegetative growth stage. NIL1913, a near isogenic line possesses CTB4a locus from KUNMINGXIAOBAIGU (KMXBG) in the background of Towada, showed stronger cold tolerance than Towada. Meanwhile, authors have confirmed the cold tolerance function of CTB4a through analyzing the phenotype of CTB4a over-expression transgenic lines, RNAi transgenic lines and loss of function mutant under cold stress conditions.

Furthermore, the authors identified 9 SNPs and 1 Indel in CTB4a promoter region, and 6 SNPs and 1 Indel in CTB4a coding regions between KMXBG and Towada. The SNPs in CTB4a promoter region were proved to be responsible for the different cold tolerance of two parental varieties. Phylogenetic analysis indicated that CTB4a did not directly originate from *O. rufipogon*, but generated from nature variation in cooler environment and retained by artificial selection.

Finally, the authors found that CTB4a encoded a conserved leucine-rich repeat receptor-like protein kinase which could influence the ATP balance through interacting with AtpB, a beta subunit of ATP synthase. This work reveals a QTL locus of cold tolerance at the booting and vegetative stage, which may improve the cold tolerance and yield of cultivated rice in cold rice growing zones.

It is an interesting story and important work. Although the tolerance at the booting stage is a technical difficulty for investigating, and a considerable work has been done.

Unfortunately, the current data are not enough to support their conclusions. For example, they showed the protein AtpB was interacted with CTB4a in biochemical and cell biological level. But there is no direct genetic evidence to show that AtpB confers cold tolerance on rice. It is a critical issue for the main conclusion in the manuscript. Additionally, there are a lot of questions in the manuscript as the followings. Therefore, the reviewer cannot recommend the manuscript of current version is published in the journal of nature communications before the manuscript is major revised and the questions are clarified.

1. The authors delimited the CTB4a-containing DNA fragment to a 56.8 kb region between marker STS5 and STS12. Nevertheless, in the Fig. 1c, recombinant C24-8 with stronger cold tolerance suggested that the CTB4a-containing DNA fragment was located at the region which didn't contain ORF4 between marker STS5 and RM5414. It seems to against the main conclusion. Please explaining for it.

2. The manuscript has proved that the truncated CTB4a could interact with AtpB. But this can't reflect reality in vivo. The authors must give evidences to show whether the full length of CTB4a interact with AtpB.

3. Whether CTB4a, which is a predicted protein kinase, can phosphorylate AtpB? If not, what's the biological significances of their interaction?

4. The manuscript suggested that CTB4a was selected by artificial and natural selection through polymorphism and geographic distribution of nine haplotypes, however the evidences were not enough to get this conclusion. Tajima's D values and MLHKA test, which can suggest whether the CTB4a has been selected or not, should be provided. In addition,

the manuscript didn't describe the methods of phylogenetic analysis in the METHODS.

5. The authors wanted to conclude a dosage effect of CTB4a on cold tolerance through the seed setting rate at the booting stage. However, the expression patterns in the transgenic lines did not match to the seed setting rate, such as, in fig. 2h-j, the seed setting rate of the line RNAi-61 with the lowest expression level of CTB4a was higher than the line RNAi-104 with the highest expression level of CTB4a. How to explain?

6. In the Fig. 2b, why the seed setting rate of C11 and C12 was not rescued to NIL1913 level? Was the expression level of CTB4a in C11 and C12 lower than NIL1913?

7. In this manuscript, the authors compared the cold tolerance between K-OE and wild-type rice or between T-OE and wild-type rice separately, but the most important comparison between K-OE and T-OE was not done. In fact, only this comparison could suggest whether the SNPs or indel in CDS of ORF4 between KMXBG and Towada were responsible for their different cold response.

8. In Supplemental Fig. 1, the authors should show the plants subjected to cold stress in deep water. Meanwhile the development stage of treated plants should be described in detail.

9. In the supplemental Fig. 10, it is not appropriate to describe the wild type rice (WT) as CK. In subgraph a, it shows that the number of pollen in K-OE is much fewer than WT, even the number of fertile pollen in K-OE is fewer than WT, this result can't demonstrate CTB4a confers cold tolerance with increased seed setting via its effect on pollen fertility.

10. In the supplementary 13a, it seems like that the cold tolerance of C11 is weaker than C12, but why the statistic result is opposite?

11. Is there any difference in seed setting rate among different transgenic plants under normal growth condition?

12. There are no section headings and subheadings in the manuscript. And a discussion for their data and clear conclusion is lacking in the manuscript.

13. In this manuscript, the specific subgraph of supplementary Fig. is not indicated when describe the specific result, such as, "Supplementary Fig. 1a and b" must be indicated in line 70.

14. In the manuscript, line 57 cited reference 2, but "low temperature" is not mentioned in this paper.

15. In the line 85, the full name of CS-HAA is mismatched. In the line 193, there is no full name for Trj.

16. The description in line 98 and 99 isn't matched with mentioned subgraph.

17. There are some mistakes with upper and lower case in references, such as reference 10 and 20.

18. In some Fig. legends, the bar lengths are not given such as, Fig. 1a?

19. In the Fig. 1, c and d should be one integrated subgraph, and detailed SNP information must be shown in the picture

20. The legend of Fig. 2 is not being organized.

21. What are the different colors indicating in supplemental Fig. 3?

22. Why supplementary Fig. 1a and 1b are the same? It is not matched to their legend.

23. The supplemental Fig. 4a is not concise enough, the authors can give the information of different segments directly on picture.

24. In the supplemental Fig. 8a, the gene symbol but not mutant number should be given.

Reviewer #2 (Remarks to the Author):

The MS by Zhang et al reports the gene CTB4a, encoding a conserved leucine-rich repeat receptor-like protein kinase, confers cold tolerance in rice. Different CTB4a alleles confer distinct levels of cold tolerance and variation in the CTB4a promoter region has been selected on the basis of temperature. Tej-Hap-KMXBG was retained by artificial selection during temperate japonica evolution. CTB4a interacts with AtpB, a beta subunit of ATP synthase, thus mediates ATP supply and photosynthetic efficiency under cold stress. Finally the yield can dramatically increase in cold rice cultivated zone.

Basically the founding is novel. The data and presentation are well organized.

There are major concerns as following that need to be addressed before the MS could be considered to be published in Nature Communications.

1. The authors claimed that CTB4a confers cold tolerance with increased seed setting via its effect on pollen fertility, but with no apparent effect in female fertility. They should illustrate the internal connection between the gene function and pollen fertility.
2. Interaction between CTB4aKD 214 and AtpB in vitro was confirmed by a GST pull down assay and in vivo by a co-immunoprecipitation (Co-IP) assay. However what is the connection between the interaction and cold tolerance?
3. The authors found that ATP synthase activity and ATP content, as well as photosynthetic rate, were significantly higher in NIL and the over-expression lines, but were significantly lower in *ctb4a* compared to HY at the booting stage under cold stress. Then, does the ATP synthase activity, or photosynthetic rate contribute to cold tolerance traits? Experimental data and explanation should be given to draw such further conclusions.
4. Does the interaction between CTB4a and AtpB affect ATP synthase activity, or photosynthetic rate? If this is the case, how can the interaction affect the ATP synthase activity?
5. Do the SNPs in the coding region affect the interaction between CTB4a and AtpB?

Reviewer #3 (Remarks to the Author):

This paper on the CTB4a gene in rice represents an important discovery in the understanding of adaptation of rice and other crops to extreme temperatures, especially to low temperature. Tolerance to cold at the reproductive stage is extremely difficult to study because of the large variations in temperature that occur in rice fields. The authors have used three different screens to measure cold tolerance in the plant materials they have studied: (a) using colder water to create a low-temperature stress on the plants, (b) low air-temperature in a phytotron, and natural conditions in a high-elevation environment where low-temperature stress occurs on a daily basis. Having a reliable location to measure the low temperature response allowed the authors to study a larger number of genotypes and this was necessary for the application of map-based cloning of the cold tolerance QTL.

The authors have done exhaustive work to delimit the gene candidate region to four open reading frames. They have also shown that the CTB4a gene which is in this region is the gene that confers cold tolerance in the most tolerant japonica rice cultivars grown in China. Even though this gene is already present in the best varieties, understanding its role in controlling this trait represents an advance in our knowledge of stress tolerance and will be useful to making further improvements in cold tolerance breeding.

In my opinion this study represents a significant advance in our knowledge and should be published in Nature Communications. I have a number of questions which I think need clarification before full acceptance of the paper. These are outlined below.

1. The authors use the term "thermophilic" to describe rice, but I have not heard this before. In fact, I think as a C3 species rice is not really "heat-loving" and tends to yield higher where temperatures are moderate. I think the term "cold-sensitive" would be more accurate.
2. In the figures showing the transgenic results, the control is always the susceptible parent. I wonder if there were any transgenics that did not express the CTB4a gene and could serve as an additional control for determining that the expression of the gene was the major cause of tolerance.
3. In p.4 lines 4-6 it mentions that there were 4 open reading frames but I could not find mention of what genes ORF1-3 were. Why was only ORF4 studied?
4. On my copy Fig.1a was very difficult to see and I wonder how useful it is.
5. In Figure 1c it seems that recombinant C24-8 has the cold tolerance trait but it might be missing the ORF4 region. This needs clarification. Also, the authors should clearly state what the different bar colors indicate (also in Suppl. Fig. 3).
6. Suppl. Fig. 1: the a and b figures seem to be identical.
7. Suppl. Fig. 5: the caption title seems to refer only to 5a (NIL 1913). 5b involved studying transgenics with different promoters.
8. Based on supplement table 2 it seems all the HAP1 varieties are grown in China but only 4 out of 28 were from other countries. Do the authors have any information on the adaptation of varieties in the extreme norther areas (e.g. Hokkaido, Vladivostok, California) which are also known to be cold tolerant? If they have a different haplotype this may represent a different mechanism of tolerance.
9. I think the results on the action of the gene, especially its interaction with an ATP synthase, is interesting. This is associated with increased ATP levels and increased photosynthesis. However, it is not clear to me that we could ascribe the effect of the gene in grain filling just to photosynthesis, especially when it seems that pollination is the major limiting factor, and photosynthesis rate is rarely associated with higher yield. I think this explanation is still tentative.
10. Some additional comments and suggestions are on the text.

Reviewer #1

The manuscript titled "Natural Variation in *CTB4a* Enhances Rice Adaptation to Cold Habitat" by authors, Zhang et al. have explored and cloned a QTL, *CTB4a*, which conferring cold tolerance at both booting stage and vegetative growth stage. NIL1913, a near isogenic line possesses *CTB4a* locus from KUNMINGXIAOBAIGU (KMXBG) in the background of Towada, showed stronger cold tolerance than Towada. Meanwhile, authors have confirmed the cold tolerance function of *CTB4a* through analyzing the phenotype of *CTB4a* over-expression transgenic lines, RNAi transgenic lines and loss of function mutant under cold stress conditions.

Furthermore, the authors identified 9 SNPs and 1 Indel in *CTB4a* promoter region, and 6 SNPs and 1 Indel in *CTB4a* coding regions between KMXBG and Towada. The SNPs in *CTB4a* promoter region were proved to be responsible for the different cold tolerance of two parental varieties. Phylogenetic analysis indicated that *CTB4a* did not directly originate from *O. rufipogon*, but generated from nature variation in cooler environment and retained by artificial selection.

Finally, the authors found that *CTB4a* encoded a conserved leucine-rich repeat receptor-like protein kinase which could influence the ATP balance through interacting with AtpB, a beta subunit of ATP synthase. This work reveals a QTL locus of cold tolerance at the booting and vegetative stage, which may improve the cold tolerance and yield of cultivated rice in cold rice growing zones.

It is an interesting story and important work. Although the tolerance at the booting stage is a technical difficulty for investigating, and a considerable work has been done. Unfortunately, the current data are not enough to support their conclusions. For example, they showed the protein AtpB was interacted with *CTB4a* in biochemical and cell biological level. But there is no direct genetic evidence to show that AtpB confers cold tolerance on rice. It is a critical issue for the main conclusion in the manuscript. Additionally, there are a lot of questions in the manuscript as the followings. Therefore, the reviewer cannot recommend the manuscript of current version is published in the journal of nature communications before the manuscript is major revised and the questions are clarified.

Answer: For the genetic evidences of *AtpB* conferring cold tolerance at the booting stage, we have obtained the *AtpB* overexpression transgenic plants and evaluated the cold tolerance of transgenic lines at the booting stage in the summer season of 2016. We found that the *AtpB* overexpression lines, *AtpB*-OE3 and *AtpB*-OE24, exhibited significant increased seed setting under CS-PT at the booting stage as shown in **Fig. 6g and Supplementary Figure 22**. In addition, to further investigate the function of *AtpB* in cold tolerance at booting stage, two mutant lines (*PFG_1B-20640.R* and *PFG_2A-10299.R*) were purchased from the Korean Rice Mutant Center, and researches for validating the detail role of *AtpB* in rice remain to be studied in our future work.

1. The authors delimited the *CTB4a*-containing DNA fragment to a 56.8 kb region between marker STS5 and STS12. Nevertheless, in the Fig. 1c, recombinant C24-8 with stronger cold tolerance suggested that the *CTB4a*-containing DNA fragment was located at the region which didn't contain ORF4 between marker STS5 and RM5414. It seems to against the main conclusion. Please explaining for it.

Answer: Sorry for the mistake, and it has been corrected in the revised manuscript as shown in **Fig. 1e**.

2. The manuscript has proved that the truncated *CTB4a* could interact with AtpB. But this can't reflect reality *in vivo*. The authors must give evidences to show whether the full length of *CTB4a* interact with

AtpB.

Answer: Thanks for the valuable comments. We conducted a bimolecular fluorescence complementation (BiFC) assay and found that both the full length of CTB4a^{KMXBG} and CTB4a^{Towada} interacted with AtpB *in vivo* as shown in **Fig. 5d**.

3. Whether CTB4a, which is a predicted protein kinase, can phosphorylate AtpB? If not, what's the biological significances of their interaction?

Answer: *CTB4a* encodes a protein kinase and we found that CTB4a exhibited autophosphorylation activity. However, as shown in **supplementary Fig.23**, we did not find that CTB4a can phosphorylate AtpB *in vitro*. Many plant RLKs need to be activated for its basal kinase function in plant cells. So it is most likely that AtpB might be phosphorylated by CTB4a *in vivo*. Additionally, it also might be the case that CTB4a interacts with AtpB to enhance the phosphorylation of AtpB by other kinases, thus affecting ATP synthase activity. Alternatively, CTB4a may regulate the assembly of ATP synthase complex under cold stress conditions. These hypotheses remain to be verified in the further studies. The detail reasons and significance of their interaction has been discussed in discussion section as shown in **p11-12 line 301-313**.

In addition, our results showed that exogenous applied of ATP could partly rescued the cold sensitivity of Towada, and *CTB4a* seemed to mediate ATP supply for reinforcing the plant resistance to cold stress.

4. The manuscript suggested that CTB4a was selected by artificial and natural selection through polymorphism and geographic distribution of nine haplotypes, however the evidences were not enough to get this conclusion. Tajima's D values and MLHKA test, which can suggest whether the CTB4a has been selected or not, should be provided. In addition, the manuscript didn't describe the methods of phylogenetic analysis in the METHODS.

Answer: We have calculated the Tajima's *D* values and conducted MLHKA test, the results were shown in **Supplemental Table 3**, which were consistent with the nucleotide diversity analysis. The method of phylogenetic analysis has been provided in **p17 line 447**.

5. The authors wanted to conclude a dosage effect of CTB4a on cold tolerance through the seed setting rate at the booting stage. However, the expression patterns in the transgenic lines did not matched to the seed setting rate, such as, in fig. 2h-j, the seed setting rate of the line RNAi-61 with the lowest expression level of CTB4a was higher than the line RNAi-104 with the highest expression level of CTB4a. How to explain?

Answer: The expression level of *CTB4a* in the panicles of homozygous RNAi lines were checked again in this season and were provided as shown in **Supplementary Fig. 6c**. Moreover, the RNAi lines were evaluated again under CS-HAA in 2016 summer season and the results were added as shown in **Fig. 2e**. Additionally, we found that the cold tolerance of different overexpression lines had a good correlation with the expression level of *CTB4a* as shown in **Supplementary Fig. 9**. These results collectively indicate the dosage effect of *CTB4a* on cold tolerance.

6. In the Fig. 2b, why the seed setting rate of C11 and C12 was not rescued to NIL1913 level? Was the expression level of CTB4a in C11 and C12 lower than NIL1913?

Answer: The expression level of *CTB4a* in C11 and C12 was not lower than NIL1913 as shown in **Supplementary Fig. 6a**. The seed setting rate of C11 and C12 was not rescued to NIL1913 level under CS-HAA conditions. It may be due to the complex and varied natural environment conditions, unlike the accurate conditions under CS-DW and CS-PT.

7. In this manuscript, the authors compared the cold tolerance between K-OE and wild-type rice or between T-OE and wild-type rice separately, but the most important comparison between K-OE and T-OE was not do. In fact, only this comparison could suggest whether the SNPs or indel in CDS of ORF4 between KMXBG and Towada were responsible for their different cold response.

Answer: The comparisons between K-OE and T-OE were provided as shown in **Fig. 2c**. Most of the seed setting of K-OE and T-OE lines showed no significant difference in three kinds of conditions.

8. In Supplemental Fig. 1, the authors should show the plants subjected to cold stress in deep water. Meanwhile the development stage of treated plants should be described in detail.

Answer: The plants subjected to cold stress in deep water were provided as shown in **Supplementary Fig. 1b and c**. The panicles at the booting stage specifically refers to the meiosis stage of the pollen mother cell and had been described in detail in the legend of **Supplementary Fig. 1c** as shown in the **p2 line 32** of supplementary information.

9. In the supplemental Fig. 10 (see supplemental Fig. 12 in revised manuscript), it is not appropriate to describe the wild type rice (WT) as CK. In subgraph a, it shows that the number of pollen in K-OE is much fewer than WT, even the number of fertile pollen in K-OE is fewer than WT, this result can't demonstrate *CTB4a* confers cold tolerance with increased seed setting via its effect on pollen fertility.

Answer: We have changed the description of "CK" to "Towada" as shown in **Supplementary Fig. 12**. In subgraph a, we calculated the ratio of fertile pollen but not the pollen number. In fact, the total pollen number showed no significant difference among them. The pictures showing that the number of pollen in K-OE is much fewer than WT was not representative and had been changed. Additionally, we added the statistical results of distorted pollen chamber as shown in **Supplementary Fig. 11c**.

10. In the supplementary 13a (see supplemental Fig. 14a in revised manuscript), it seems like that the cold tolerance of C11 is weaker than C12, but why the statistic result is opposite?

Answer: At least five pots for each material were used to evaluate the cold tolerance at the seedling stage, and in fact the survival rates of C11 and C12 showed no significant difference. The picture of C11 changed in **Supplementary Fig. 14a** is more representative than before.

11. Is there any difference in seed setting rate among different transgenic plants under normal growth condition?

Answer: The seed setting of different complementation and overexpression lines showed no significant difference under normal growth condition, but not the RNAi lines and mutant. In the manuscript, we used relative value for CS-DW and CS-PT. Relative seed setting is the seed setting rate under cold stress divided by the seed setting under normal conditions.

12. There are no section headings and subheadings in the manuscript. And a discussion for their data and clear conclusion is lack in the manuscript.

Answer: The last version of our manuscript submitted to *Nature Communications* was transferred from *Nature Genetics* directly and it was prepared as a letter format. Now we had changed it into article format and the subheadings, a discussion, a conclusion were added in the manuscript.

13. In this manuscript, the specific subgraph of supplementary Fig. is not indicated when describe the specific result, such as, “Supplementary Fig. 1a and b” must be indicated in line 70.

Answer: We have carefully checked and add the indication of some subgraph of supplementary Fig.

14. In the manuscript, line 57 cited reference 2, but “low temperature” is not mentioned in this paper.

Answer: It should be cited as reference 46 as shown in **p21 line 609** and the new reference 2 has been added.

15. In the line 85, the full name of CS-HAA is mismatched. In the line 193, there is no full name for Trj.

Answer: The location of full name of CS-HAA had been changed to match the description as shown in the **p4 line99**, and the full name for *Trj* was added as shown in **p9 line232**.

16. The description in line 98 and 99 isn't matched with mentioned subgraph.

Answer: Sorry for the mistakes and these mistakes have been corrected.

17. There are some mistakes with upper and lower case in references, such as reference 10 and 20.

Answer: These mistakes have been corrected as shown in the references.

18. In some Fig. legends, the bar lengths are not given such as, Fig. 1a?

Answer: These mistakes have been corrected.

19. In the Fig. 1, c and d (see Fig. 1e in revised manuscript) should be one integrated subgraph, and detailed SNP information must be showed in the picture

Answer: These two parts have been combined and the detailed SNP information had been added as shown in **Fig.1e**.

20. The legend of Fig. 2 is not being organized.

Answer: The legend of six figures have been organized again as shown in the manuscript

21. What are the different colors indicating in supplemental Fig. 3?

Answer: The black box indicates homozygous KMXBG DNA, the white box indicates homozygous Towada DNA and gray box indicates the heterozygous DNA. This information has been added as shown in **Supplemental Fig. 3**

22. Why supplementary Fig. 1a and 1b are the same? It is not matched to their legend.

Answer: This kind of mistake has been corrected.

23. The supplemental Fig. 4a is not concise enough, the authors can give the information of different segments directly on picture.

Answer: As suggested, a Schematic diagram of CTB4a protein was added as shown in **Supplemental**

Fig. 4a.

24. In the supplemental Fig. 8a ((see supplemental Fig. 10a in revised manuscript)), the gene symbol but not mutant number should be given.

Answer: The mutant number was replaced with gene symbol as shown in **Supplemental Fig. 10a.**

Reviewer #2

The MS by Zhang et al reports the gene CTB4a, encoding a conserved leucine-rich repeat receptor-like protein kinase, confers cold tolerance in rice. Different CTB4a alleles confer distinct levels of cold tolerance and variation in the CTB4a promoter region has been selected on the basis of temperature. Tej-Hap-KMXBG was retained by artificial selection during temperate japonica evolution. CTB4a interacts with AtpB, a beta subunit of ATP synthase, thus mediates ATP supply and photosynthetic efficiency under cold stress. Finally the yield can dramatically increase in cold rice cultivated zone. Basically the founding is novel. The data and presentation are well organized. There are major concerns as following that need to be addressed before the MS could be considered to be published in Nature Communications.

1. The authors claimed that CTB4a confers cold tolerance with increased seed setting via its effect on pollen fertility, but with no apparent effect in female fertility. They should illustrate the internal connection between the gene function and pollen fertility.

Answer: Thanks for the valuable comment, and we have added it as shown in discussion section in p11 line 289-300:

“Low temperature leads to a series of changes at the molecular and cellular levels as well as physiological and biochemical levels in plant cells. The plants need energy for reinforcing its resistance to cold stress, among which ATP is one of the important energy sources³⁵⁻³⁸. Our results demonstrated that up-regulation of *CTB4a* increased ATP synthase activity and ATP content in NIL1913 and overexpression lines under cold stress at the booting stage (**Fig. 6a-d**). And higher pollen fertility was observed in NIL1913 and *CTB4a* overexpression lines than Towada under cold stress (**Fig. 1c, d and Supplementary Fig. 12**). Moreover, the application of exogenous ATP increased seed setting of Towada under cold stress (**Fig. 6e**). So it is likely that the low expression of *CTB4a* and reduced ATP content in Towada led to less energy for enhancing its cold tolerance, which gives rise to reduced pollen fertility and decreased seed setting. As previously reported, the shortage in ATP content reduces grain productivity^{39, 40}. We finally found that NIL1913 and *CTB4a* overexpression lines exhibited improved grain yield under CS-HAA (**Fig. 6f**).”

2 AND 4. Interaction between CTB4a^{KD} 214 and AtpB in vitro was confirmed by a GST pull down assay and in vivo by a co-immunoprecipitation (Co-IP) assay. However what is the connection between the interaction and cold tolerance? Does the interaction between CTB4a and AtpB affect ATP synthase activity, or photosynthetic rate? If this is the case, how can the interaction affect the ATP synthase activity?

Answer: We found that CTB4a interacted with AtpB *in vivo* and *in vitro* as shown in **Fig. 5**. To further investigate the function of AtpB for cold tolerance, we obtained AtpB overexpression lines and found that AtpB-OE3 and AtpB-OE24 showed significantly increased seed setting than that of Nip under cold stress at the booting stage as shown in **Fig. 6g and Supplementary Fig. 22**. Moreover, ATP synthase activity and ATP content as well as photosynthetic rate were significantly higher in NIL1913 and the overexpression lines than in Towada. And exogenous application of ATP partly rescued the cold sensitivity of Towada. These results collectively indicate that CTB4a interacts with AtpB and likely mediate ATP synthesis, ensuring an energy supply, thereby enhancing seed setting rates and increasing yield in cold rice growing areas.

CTB4a encodes a protein kinase and we found that CTB4a exhibited autophosphorylation activity. However, as shown in **supplementary Fig.23**, we did not find that CTB4a can phosphorylate AtpB *in vitro*. Many plant RLKs need to be activated for its basal kinase function in plant cells. So it is most likely that AtpB might be phosphorylated by CTB4a *in vivo*. Additionally, it also might be the case that CTB4a interacts with AtpB to enhance the phosphorylation of AtpB by other kinases, thus affecting ATP synthase activity. Alternatively, CTB4a may regulate the assembly of ATP synthase complex under cold stress conditions. These hypotheses remain to be verified in the further studies. The detail reasons and significance of their interaction has been discussed in discussion section as shown in **p11-12 line 301-313**.

3. The authors found that ATP synthase activity and ATP content, as well as photosynthetic rate, were significantly higher in NIL and the over-expression lines, but were significantly lower in *ctb4a* compared to HY at the booting stage under cold stress. Then, does the ATP synthase activity, or photosynthetic rate contribute to cold tolerance traits? Experimental data and explanation should be given to draw such further conclusions.

Answer: As mentioned in the answer 2 and 4, the genetic evidence from *AtpB* overexpression lines confirm the function of *AtpB* for cold tolerance. Moreover, exogenous applied of ATP could partly rescued the cold sensitivity of Towada. Taken together, these results imply that the increase of ATP can improve the cold tolerance in rice at the booting stage.

5. Do the SNPs in the coding region affect the interaction between CTB4a and AtpB?

Answer: We conducted a bimolecular fluorescence complementation (BiFC) assay and found that both the full length of CTB4a^{KMXBG} and CTB4a^{Towada} interacted with AtpB *in vivo* as shown in **Fig. 5d**.

Reviewer #3

This paper on the *CTB4a* gene in rice represents an important discovery in the understanding of adaptation of rice and other crops to extreme temperatures, especially to low temperature. Tolerance to cold at the reproductive stage is extremely difficult to study because of the large variations in temperature that occur in rice fields. The authors have used three different screens to measure cold tolerance in the plant materials they have studied: (a) using colder water to create a low-temperature stress on the plants, (b) low air-temperature in a phytotron, and natural conditions in a high-elevation environment where low-temperature stress occurs on a daily basis. Having a reliable location to measure the low temperature response allowed the authors to study a larger number of genotypes and this was necessary for the application of map-based cloning of the cold tolerance QTL. The authors have done exhaustive work to delimit the gene candidate region to four open reading frames. They have also shown that the *CTB4a* gene which is in this region is the gene that confers cold tolerance in the most tolerant japonica rice cultivars grown in China. Even though this gene is already present in the best varieties, understanding its role in controlling this trait represents an advance in our knowledge of stress tolerance and will be useful to making further improvements in cold tolerance breeding.

In my opinion this study represents a significant advance in our knowledge and should be published in Nature Communications. I have a number of questions which I think need clarification before full acceptance of the paper. These are outlined below.

1. The authors use the term “thermophilic” to describe rice, but I have not heard this before. In fact, I think as a C3 species rice is not really “heat-loving” and tends to yield higher where temperatures are moderate. I think the term “cold-sensitive” would be more accurate.

Answer: Thanks for the valuable comment, and we have changed the term “thermophilic” to “cold-sensitive” or “cold-tolerant” in the manuscript.

2. In the figures showing the transgenic results, the control is always the susceptible parent. I wonder if there were any transgenics that did not express the *CTB4a* gene and could serve as an additional control for determining that the expression of the gene was the major cause of tolerance.

Answer: Except for Towada, we had other two controls, namely K-NT and T-NT as shown in **Supplementary Figure 8a** and **Supplementary Figure 9**. K-NT and T-NT indicates the non-transgenic plants for K-OE and T-OE vector respectively. We found that the seed setting among Towada, K-NT and T-NT showed no significant difference under cold stress.

3. In p.4 lines 4-6 it mentions that there were 4 open reading frames but I could not find mention of what genes ORF1-3 were. Why was only ORF4 studied?

Answer: There were 4 open reading frames in the mapped region. Firstly, we analyzed the nucleotide sequence of them and found that ORF2 and ORF3 were identical between two parents. As for ORF1, only one rare SNP was found in the coding region among different cultivars. 9 SNPs and 1 Indel in ORF4 promoter region, and 6 SNPs and 1 Indel in ORF4 coding region were found between two parents. Haplotype analysis of ORF4 showed that the KMXBG haplotype exhibited stronger cold tolerance than

others. What's more, ORF4 could be higher induced by cold stress in NIL1913 than Towada. Based on this, ORF4 were selected as candidate gene for analysis.

4. On my copy Fig.1a was very difficult to see and I wonder how useful it is.

Answer: **In this version, the clearer photos were provided in the Fig.1a.**

5. In Figure 1c it seems that recombinant C24-8 has the cold tolerance trait but it might be missing the ORF4 region. This needs clarification. Also, the authors should clearly state what the different bar colors indicate (also in Suppl. Fig. 3).

Answer: **Sorry for the mistake and it has been corrected in the revised manuscript as shown in Fig. 1e, and the means of different bar colors have been added as shown in Fig. 1e and Supplementary Figure 3b.**

6. Suppl. Fig. 1: the a and b figures seem to be identical.

Answer: **Sorry for the mistake and this kind of mistake has been corrected.**

7. Suppl. Fig. 5: the caption title seems to refer only to 5a (NIL 1913). 5b involved studying transgenics with different promoters.

Answer: **This figure has been divided into Supplementary Figure 5 and Supplementary Figure 7 in this version, and the caption titles have been added.**

8. Based on supplement table 2 it seems all the HAP1 varieties are grown in China but only 4 out of 28 were from other countries. Do the authors have any information on the adaptation of varieties in the extreme norther areas (e.g. Hokkaido, Vladivostok, California) which are also known to be cold tolerant? If they have a different haplotype this may represent a different mechanism of tolerance.

Answer: **Regrettably, we have screened all the germplasm in our laboratory and found that only two varieties (see Supplementary table2, No.123 and 135) were from Russia, and both of them belonged to the cold-tolerant haplotype Hap1.**

9. I think the results on the action of the gene, especially its interaction with an ATP synthase, is interesting. This is associated with increased ATP levels and increased photosynthesis. However, it is not clear to me that we could ascribe the effect of the gene in grain filling just to photosynthesis, especially when it seems that pollination is the major limiting factor, and photosynthesis rate is rarely associated with higher yield. I think this explanation is still tentative.

Answer: **According to the valuable comment, we have changed our description. For example, as shown in p10 line 272 in results section "These results collectively indicate that CTB4a interacts with AtpB and likely mediates ATP synthesis, ensuring an energy supply, thereby enhancing seed setting rates and increasing yield in cold rice growing areas." and in p11 line 298 in discussion section "As previously reported, the shortage in ATP content reduces grain productivity^{39, 40}. We found that NIL1913 and CTB4a overexpression lines exhibited improved grain yield under CS-HAA (Fig. 6g)."**

10. Some additional comments and suggestions are on the text

Answer: Thanks for the valuable comments, and most of them have been adopted as shown in the manuscript.

REVIEWERS' COMMENTS:

Reviewer #1 (Remarks to the Author):

In the revised manuscript "Natural Variation in CTB4a Enhances Rice Adaptation to Cold Habitat", all necessary experiments had been done by the authors, and the questions were well answered. But, the answer of the question about recombinant C24-8 seems like not sufficiently rigorous, the authors should provide additional evidences to prove the chromosome structure of recombinant C24-8 in the revised manuscript but not the former one is true. The precise localization of CTB4a is the foundation of this work, so the confirmation of this result is vital important.

Reviewer #2 (Remarks to the Author):

The revised manuscript "Natural Variation in CTB4a Enhances Rice Adaptation to Cold Habitat" by Zhang et al. has addressed most of the concerns from this reviewer.

Although the mechanism of CTB4a has not been clarified to a more satisfactory extent for being considered to be published in Nature Communications, there are more detailed explanations for the innovative discoveries than in the previous version.

The manuscript of current version could be recommendable to be published in Nature Communications.

Reviewer #3 (Remarks to the Author):

As mentioned in my first review, I believe this is a significant study and provides important information on cold tolerance, which is very difficult to measure in realistic conditions. The authors have answered my initial questions, and I support publication of the manuscript (subject to editorial improvements in grammar, etc.). I have two suggestions for improving it:

L88: "This work uncovers a novel gene for cold tolerance and its probability in breeding practice." I think the statement is a vague and I am not sure what is meant by probability in breeding.

L201- "Also, 18 wild rice accessions were sequenced and although we found similar diversity to that in cultivated rice accessions, no haplotypes identical to Hap-KMXBG or Hap-Towada were found (Fig. 3a). These results show that cold tolerant haplotype of CTB4a originates from and is only present in japonica."

I think the statement is too strong because we cannot rule out that the gene was present in some ancestral wild strains or landraces which are not available or have not been studied.

Reviewer #1 (Remarks to the Author):

In the revised manuscript "Natural Variation in CTB4a Enhances Rice Adaptation to Cold Habitat", all necessary experiments had been done by the authors, and the questions were well answered. But, the answer of the question about recombinant C24-8 seems like not sufficiently rigorous, the authors should provide additional evidences to prove the chromosome structure of recombinant C24-8 in the revised manuscript but not the former one is true. The precise localization of CTB4a is the foundation of this work, so the confirmation of this result is vital important.

Answer: Thanks for the comments, the following picture showed the genotype of C24-8 by DNA sequencing at maker RM5414 (A) and STS12 (B). The results indicated that the C24-8 consisted of KMXBG-genotype at RM5414 and Towada-genotype at STS12. So the localization of CTB4a should be between STS5 and STS12. It was not the mistake made during genotyping of the plants, but during drawing the schematic diagram.

Fig. Genotyping of recombinant C24-8 by DNA sequencing. The red arrow in (A) and box in (B) indicated the differences.

Reviewer #2 (Remarks to the Author):

The revised manuscript "Natural Variation in CTB4a Enhances Rice Adaptation to Cold Habitat" by Zhang et al. has addressed most of the concerns from this reviewer.

Although the mechanism of CTB4a has not been clarified to a more satisfactory extent for being considered to be published in Nature Communications, there are more detailed explanations for the innovative discoveries than in the previous version.

The manuscript of current version could be recommendable to be published in Nature Communications.

Answer: Thanks for your review of our manuscript, and many valuable comments were raised.

Reviewer #3 (Remarks to the Author):

As mentioned in my first review, I believe this is a significant study and provides important information on cold tolerance, which is very difficult to measure in realistic conditions. The

authors have answered my initial questions, and I support publication of the manuscript (subject to editorial improvements in grammar, etc.). I have two suggestions for improving it:

L88: "This work uncovers a novel gene for cold tolerance and its probability in breeding practice." I think the statement is a vague and I am not sure what is meant by probability in breeding.

Answer: Thanks for the valuable comment, The editor has revised it as "This work uncovers a novel gene for cold tolerance with potential value in breeding" showed in line 92, and "New cultivars designed on the basis of our study could test whether CTB4a can confer cold tolerance in other crops in the future" showed in line 351, and so on. We agree with these more appropriate descriptions raised by editor.

L201- "Also, 18 wild rice accessions were sequenced and although we found similar diversity to that in cultivated rice accessions, no haplotypes identical to Hap-KMXBG or Hap-Towada were found (Fig. 3a). These results show that cold tolerant haplotype of CTB4a originates from and is only present in japonica."

I think the statement is too strong because we cannot rule out that the gene was present in some ancestral wild strains or landraces which are not available or have not been studied.

Answer: Thanks for the valuable suggestion, we have revised it as "These results show that cold tolerant haplotype of CTB4a originates from japonica and is probably just present in japonica." showed in L211.